# $\mathbb{X}$-Sample Contrastive Loss: Improving Contrastive Learning with Sample Similarity Graphs

**Vlad Sobal**[1,2]    **Mark Ibrahim**[1]    **Randall Balestriero**[3]    **Vivien Cabannes**[1]
**Diane Bouchacourt**[1]    **Pietro Astolfi**[1]    **Kyunghyun Cho**[2,4,5]    **Yann LeCun**[1,2]
[1]Meta FAIR    [2]New York Univeristy    [3]Brown University    [4]Genentech    [5]CIFAR

## Abstract

Learning good representations involves capturing the diverse ways in which data samples relate. Contrastive loss—an objective matching related samples—underlies methods from self-supervised to multimodal learning. Contrastive losses, however, can be viewed more broadly as modifying a similarity graph to indicate how samples should relate in the embedding space. This view reveals a shortcoming in contrastive learning: the similarity graph is binary, as only one sample is the related positive sample. Crucially, similarities *across* samples are ignored. Based on this observation, we revise the standard contrastive loss to explicitly encode how a sample relates to others. We experiment with this new objective, called $\mathbb{X}$-Sample Contrastive, to train vision models based on similarities in class or text caption descriptions. Our study spans three scales: ImageNet-1k with 1 million, CC3M with 3 million, and CC12M with 12 million samples. The representations learned via our objective outperform both contrastive self-supervised and vision-language models trained on the same data across a range of tasks. When training on CC12M, we outperform CLIP by $0.6\%$ on both ImageNet and ImageNet Real. Our objective appears to work particularly well in lower-data regimes, with gains over CLIP of $17.2\%$ on ImageNet and $18.0\%$ on ImageNet Real when training with CC3M. Finally, our objective encourages the model to learn representations that separate objects from their attributes and backgrounds, with gains of 3.3-5.6% over CLIP on ImageNet9. The proposed method takes a step towards developing richer learning objectives for understanding sample relations in foundation models.

## 1 Introduction

Contrastive loss underlies methods from self-supervised learning (SSL) to multimodal learning (Radford et al., 2021; Chen et al., 2020; Oord et al., 2018). In SSL, contrastive learning encourages the model to associate a sample with another view of the sample created using hand-crafted data augmentation—this related view is the positive sample. Other samples are then pushed away as negative, unrelated samples in the models' representation space. Contrastive losses also play a crucial role in multimodal models such as CLIP (Radford et al., 2021), where the model associates an image with its text caption in representation space. Here contrastive learning designates the caption and image representations as positives while all other text-image pairs are designated as negatives.

More broadly, contrastive losses can be seen as constructing a similarity graph to indicate how samples should relate in the model's representation space (Cabannes et al., 2023; Zhang et al., 2023; HaoChen et al., 2021; Wang et al., 2023; 2022). This view reveals a shortcoming in contrastive learning: the similarity graph is binary, as only one sample is the related positive sample. Crucially, similarities across samples which contain precious signals about how aspects of one sample may relate to another, are ignored, thereby limiting the quality of the learned representations. For the example shown in fig. 1a, contrastive learning treats each image independently, without explicitly encoding similarities between the images depicting different pets. While prior work showed that data samples can be implicitly linked by multi-step connections in the augmentations graph when using contrastive learning (Wang et al., 2022; 2023), we explore how to capture such similarities explicitly by modifying the standard contrastive objective. We show that, in contrast to modeling

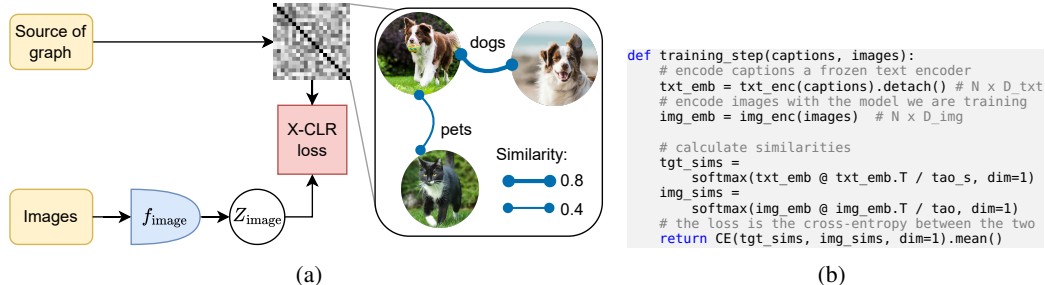

Figure 1: **a) The diagram of $\mathbb{X}$-CLR**. $\mathbb{X}$-CLR objective learns representations of images with the help of a soft relationship graph. The graph can be built based on accompanying data, e.g. taxonomy for biological data. In our experiments, we use captioned images, and build similarities based on caption similarities. **b) Python-style pseudo-code of $\mathbb{X}$-CLR with similarity based on text captions.**

those connections implicitly, explicitly incorporating similarities between visually different but semantically related samples makes representations more robust.

To account for similarities across samples, we first remove the binary negative vs. positive designations in standard contrastive loss. We introduce instead a similarity graph with continuous scalars capturing the extent to which two samples are related. Consider the example in fig. 1, where the two dog images have a high similarity while the dog and cat images have a more moderate similarity. We build our objective by incorporating the soft similarity targets into the InfoNCE objective (Oord et al., 2018) in a manner similar to objectives used for distillation (Hinton et al., 2015; Wu et al., 2023). In contrast to distillation, we do not focus on training a model using outputs of another model, but instead focus on building a meaningful similarities graph and incorporating it into the objective. *The proposed target similarity graph does not have to come from another model*, and can be inferred from any additional metadata. We experiment with this new objective, called $\mathbb{X}$-Sample Contrastive Learning ($\mathbb{X}$-CLR), by training vision models using a graph of similarities inferred from class or text caption descriptions found in common datasets. Our study spans three training dataset scales from 1 million samples with high-quality labels from ImageNet (Deng et al., 2009) to 3 and 12 million noisy image-text caption pairs from CC3M and CC12M (Sharma et al., 2018).

We find that compared to contrastive baseline methods trained on the same data, representations trained using $\mathbb{X}$-CLR outperform contrastive training on a range of tasks from standard classification to tasks involving the decomposition of objects from their attributes and backgrounds. When training on CC12M, we outperform CLIP by 0.6% on both ImageNet and ImageNet Real (Beyer et al., 2020). Furthermore, $\mathbb{X}$-CLR yields representations that separate objects from their attributes and backgrounds well, with gains of 3.4-4.9% over CLIP on ImageNet9 (Xiao et al., 2020). We also find for fine-grained disambiguation of object attributes, the quality of labels used to infer the similarity graph is much more important than the data quantity. Compared to noisier web caption data, we find $\mathbb{X}$-CLR trained on 1 million higher quality class labels outperforms representations learned via standard contrastive CLIP trained $12\times$ more data. Finally, we find $\mathbb{X}$-CLR appears to work particularly well in lower-data regimes, with gains over CLIP of 16.8% on ImageNet and 18.1% on ImageNet Real when training with CC3M. In short, we find representations learned using $\mathbb{X}$-CLR generalize better, decompose objects from their attributes and backgrounds, and are more data-efficient.

Our contributions are:

1. We revisit the graph similarity perspective of contrastive losses, revealing that standard losses encode a sparse similarity matrix that treats other possibly related samples as negatives;

2. We propose a new $\mathbb{X}$-CLR loss that explicitly accounts for soft similarities across samples;

3. We experiment with this objective across three levels of data scale from 1-12 million samples, and find that the representations learned via $\mathbb{X}$-CLR:

   (a) Generalize better on standard classification tasks with consistent gains over contrastive baselines trained on the same data. For example, when training on CC12M we outperform CLIP by 0.6% on both ImageNet and ImageNet Real.

(b) Disambiguate aspects of images such as attributes and backgrounds more reliably, with gains of 3.3-5.6% over CLIP on background robustness benchmarks for ImageNet.

(c) Finally, we find $\mathbb{X}$-CLR learns more efficiently when data is scarce, with gains of 17.2% on ImageNet and 18.0% on ImageNet Real when pretraining on the smaller 3 million sample CC3M dataset.

The proposed solution takes a step towards developing richer learning objectives for understanding sample relations in foundation models to encode richer, more generalizable representations.

## 2 RELATED WORK

**Contrastive learning** Various contrastive objectives have been proposed over the years (Chopra et al., 2005; Schroff et al., 2015). More recently, the InfoNCE objective (Oord et al., 2018) has been the most popular choice for self-supervised methods, e.g. SimCLR (Chen et al., 2020) and MoCo (He et al., 2020). InfoNCE objective has also been successfully used to learn vision-language models using CLIP (Radford et al., 2021). The basis of those objectives is to make positive pairs have similar representations, while the negatives, which typically are just all other elements in a batch, should have a different representation. In its original form, InfoNCE is binary, meaning it only works with positive and negative pairs, and does not support degrees of similarity. The positive pairs are usually two augmentations of the same sample, which makes well-tuned augmentations crucial for good performance (Ryali et al., 2021). Dwibedi et al. (2021) estimate positives using nearest neighbors in the latent space instead and therefore can use weaker augmentations, while Caron et al. (2020) use cluster assignment. A few methods have proposed modifications wherein multiple positive pairs are supported, e.g., Khosla et al. (2020) groups positive by class labels, Hoffmann et al. (2022) propose using WordNet (Fellbaum, 1998) hierarchy to define ranked positive samples, and Tian et al. (2024) uses a generative model to obtain multiple positives for the same concept. HaoChen et al. (2021) also look at contrastive learning through the lens of graphs, and propose a novel spectral objective. Wang et al. (2023) draw connections between contrastive learning and message passing on the augmentation graph, while Wang et al. (2022) show that aggressive data-augmentations like cropping can connect samples of the same class. Zhang et al. (2023) show that contrastive learning objective implicitly learns the graph in which the samples are connected via augmentations in the case of SimCLR or via captions in the case of CLIP. However, in that paradigm only visually similar samples or samples with a common caption get connected in the graph, while in our method the samples are connected based on provided graph, which can connect visually different yet semantically related samples.

**Soft targets** Using soft targets provides more learning signal to the model, possibly making it learn better and faster. This has been explored with distillation by Hinton et al. (2015). Soft targets have also been used with InfoNCE in the context of distillation in ReSSL (Zheng et al., 2021) and SCE (Denize et al., 2023), where the target cross-sample similarity comes from the teacher model. (Feng and Patras, 2023) use soft targets from self-distillation to train an image encoder with coarse labels. Similarly, Fini et al. (2023a) compute soft targets via latent clustering and apply it to semi-supervised learning. Shen et al. (2023) use patch-mixing to train ViT image encoders to model inter-sample relationships. Andonian et al. (2022) proposes to use soft targets for CLIP (Radford et al., 2021) training, and calculates the targets via self-distillation. Wu et al. (2023) use a similar objective to ours to distill the CLIP model into a smaller one. Further soft CLIP objectives are explored by Fini et al. (2023b), who apply label smoothing to obtain soft targets, and Gao et al. (2024), who estimate soft targets by comparing fine-grained image information. Finally, Huang et al. (2024) train CLIP with non-zero cross-sample similarities computed based on pre-trained uni-modal models for text and vision. In this study, we build on the work of Cabannes et al. (2023) who propose a unifying framework to view SSL and supervised learning objectives as learning with different underlying similarity graphs. We take inspiration from the soft targets literature and propose using a soft graph. Unlike distillation methods, where the soft targets typically come from a teacher model, the targets in $\mathbb{X}$-CLR can originate from any source, not necessarily another model. We try different similarity graph sources, including ones not based on the outputs of another model, and show that we can build a better graph than ones commonly used in contrastive learning (see table 2).

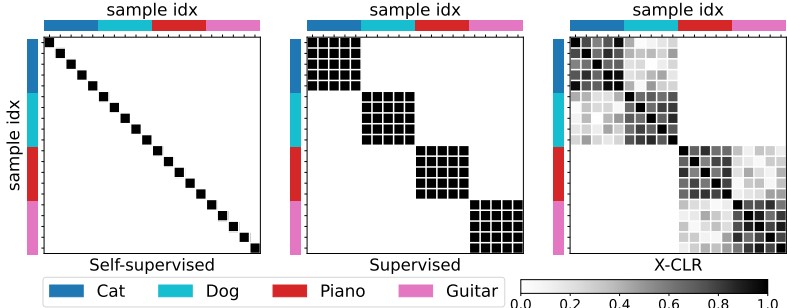

Figure 2: **Sample similarity adjacency matrices of existing methods vs. our $\mathbb{X}$-Sample Contrastive similarity loss (right).** We show pairwise similarities of 20 samples belonging to 4 classes. Similarity of 1 means the samples are identical, 0 – they are completely unrelated. In case of self-supervised learning, none of the inter-sample relationships are modelled (left). Supervised learning relies on the labels to group samples of the same class together (center). $\mathbb{X}$-CLR models inter-class relationships by associating cats with dogs and pianos with guitars.

## 3 UNDERSTANDING CONTRASTIVE LOSSES VIA SIMILARITY GRAPHS

### 3.1 X-SAMPLE GRAPHS

Throughout this study, a similarity graph denotes a graph in which the nodes represent data samples, and edges – similarity relationships. Given the number of data samples in the dataset $N$, a graph is expressed through its symmetric adjacency matrix $\boldsymbol{G} \in \mathbb{R}^{N \times N}$, the semantic relation between inputs $i$ and $j$ being encoded in the real entry $\boldsymbol{G}_{i,j}$. In fig. 2, we show graphs of different learning paradigms. SSL does not rely on labels, but on positive pairs/tuples/views generated at each epoch. Let us denote by $V$ the number of positive views generated, commonly $V = 2$ for positive pairs, and denote by $E$ the training epochs. In that case, the original $N$ input samples are transformed into $N \times V \times E$ "augmented" samples

$$\boldsymbol{X}^{(A)} \triangleq [\underbrace{\mathcal{T}(\boldsymbol{x}_1), \ldots, \mathcal{T}(\boldsymbol{x}_1)}_{\text{repeated } V \times E \text{ times}}, \ldots, \mathcal{T}(\boldsymbol{x}_N), \ldots, \mathcal{T}(\boldsymbol{x}_N)]^\top,$$

where each $\mathcal{T}$ is a random input transformation with its own random parameters. The corresponding graph is given by:

$$\boldsymbol{G}_{i,j}^{(\text{ssl})} = \mathbf{1}_{\{\lfloor i/VE \rfloor = \lfloor j/VE \rfloor\}}, \tag{1}$$

where the associated similarity graph captures if two samples were generated as augmentations of the same original input. Such graphs $\boldsymbol{G}$, as defined by eq. (1), are the ones used as targets in common SSL methods, as formalized below denoting $\boldsymbol{Z} \triangleq f_\theta(\boldsymbol{X}) \in \mathbb{R}^{N \times D}$. Here, $f_\theta$ is the encoder, and $D$ is the encoding dimension.

**Theorem 1** ((Cabannes et al., 2023)). *VICReg (Bardes et al., 2021), SimCLR (Chen et al., 2020), and BarlowTwins (Zbontar et al., 2021) losses can be expressed in terms of the graph $\boldsymbol{G}$ (1)*

$$\mathcal{L}_{\text{VIC}^2}(\boldsymbol{Z}; \boldsymbol{G}) = \|\boldsymbol{Z}\boldsymbol{Z}^T - \boldsymbol{G}\|_F^2,$$

$$\mathcal{L}_{\text{SimCLR}}(\boldsymbol{Z}; \boldsymbol{G}) = -\sum_{i,j \in [N]} \boldsymbol{G}_{i,j} \log \left( \frac{\exp(\tilde{\boldsymbol{z}}_i^\top \tilde{\boldsymbol{z}}_j)}{\sum_{k \in [N]} \exp(\tilde{\boldsymbol{z}}_i^\top \tilde{\boldsymbol{z}}_k)} \right),$$

$$\mathcal{L}_{\text{BT}}(\boldsymbol{Z}; \boldsymbol{G}) = \left\| \tilde{\boldsymbol{Z}}^\top \boldsymbol{G} \tilde{\boldsymbol{Z}} - I \right\|^2,$$

*where $\tilde{\boldsymbol{z}} \triangleq \boldsymbol{z}/\|\boldsymbol{z}\|$ and $\tilde{\boldsymbol{Z}}$ the column normalized $\boldsymbol{Z}$ so that each column has unit norm.*

In our study, we will focus on contrastive learning, i.e., SimCLR family of losses. We will demonstrate how to move away from the ad-hoc graph $\boldsymbol{G}$ from eq. (1).

### 3.2 REVISITING CONTRASTIVE LOSSES WITH SIMILARITY GRAPHS: $\mathbb{X}$-CLR

We introduce the soft cross-sample similarity to the widely used InfoNCE objective (Oord et al., 2018). We note that the proposed framework isn't necessarily limited to InfoNCE-based methods and can potentially be integrated into non-contrastive objectives such as BYOL, SimSiam, or VICReg (Grill et al., 2020; Chen and He, 2020; Bardes et al., 2021), although we leave the extensions to future work. In SimCLR (Chen et al., 2020), given a batch of $N_b$ images, each image is augmented twice, so each sample has a true positive. The $2N_b$ images are then encoded representations $z$. Then:

$$p_{i,j} = \frac{\exp(\mathrm{sim}(z_i, z_j)/\tau)}{\sum_{k=1}^{2N} \mathbf{1}_{k \neq i} \exp(\mathrm{sim}(z_i, z_k)/\tau)}$$

$$\mathcal{L}_{\mathrm{SimCLR}} = \frac{1}{2N_b} \sum_{i=1}^{2N_b} H(\mathbb{1}_{i'}, p_i)$$

where $H$ is the cross-entropy, and $\mathbb{1}_{i'}$ is the one-hot distribution where all the probability mass is assigned to the index of the positive sample corresponding to $i$, and $\mathrm{sim}$ is the cosine similarity. Intuitively, we are training the model to classify positive examples in a batch, so the similarity $p$ should be high only for the true positive. We introduce the soft objective by replacing the hard positive distribution $\mathbb{1}_{i'}$ with a distribution $s_i$. Or, in terms of graphs, we replace the graph from the eq. (1) with a soft graph where connection strengths can be any number in $[0, 1]$, and, similarly, the distribution $s_i$ and does not have to be one-hot. Considering the example of fig. 1, we want a photo of a dog to have a representation similar to that of another photo of a dog, somewhat similar to the representation of a cat photo, and different from the representation of a photo of a mug. Given that distribution $s$, we can plug it in directly:

$$\mathcal{L}_{\mathbb{X}\text{-CLR}} = \frac{1}{2N_b} \sum_{i=1}^{2N_b} H(s_i, p_i)$$

There are many possible ways to obtain this distribution $s$. We could use the meta-data associated with the dataset. In our particular case, we utilize a trained text encoder $f_{\mathrm{text}}$, and encode the text provided with each image to obtain a representation, which is then used to calculate similarity between samples $i$ and $j$ using the cosine similarity. Those pairwise similarities describe the soft graph:

$$\boldsymbol{G}_{i,j}^{(\mathrm{soft})} = \mathrm{sim}(f_{\mathrm{text}}(c_i), f_{\mathrm{text}}(c_j))$$

Where $c_i$ is the caption associated with the $i$-th sample. Note that the similarities $\boldsymbol{G}^{(\mathrm{soft})}$ do not have to come from another model, and can be inferred based on any additional information available, making this type of objective applicable to other datasets and modalities. The last step before plugging the similarities into the objective is converting them to a valid probability distribution using softmax:

$$s_{i,j} = \frac{\exp(\boldsymbol{G}_{i,j}^{(\mathrm{soft})}/\tau_s)}{\sum_{k=1}^{2N_b} \exp(\boldsymbol{G}_{i,k}^{(\mathrm{soft})}/\tau_s)}$$

$\tau_s$ is a separate hyperparameter from $\tau$ in the softmax to calculate the learned similarities. Higher values of $\tau_s$ put more weight on the 'soft' positives, while lower values in the limit recover the original SimCLR objective. For more details about this, see appendix A.9.

## 4 EXPERIMENTS

### 4.1 EXPERIMENTAL SETUP

We test $\mathbb{X}$-CLR on three datasets of varying scale: ImageNet (Deng et al., 2009) (1M), and Conceptual Captions 3M and 12M (Sharma et al., 2018). We blur faces in all datasets before training our models. We use the Sentence Transformer (Reimers and Gurevych, 2019) as the text encoder to construct similarities unless stated otherwise. For ImageNet experiments, we generate captions by using the template "a photo of a _" to generate captions out of class names. In our experiments with the conceptual captions dataset (Sharma et al., 2018), we use the captions as is. For experiments on

Table 1: **Accuracy of $\mathbb{X}$-CLR and a range of baselines on a set of classification benchmarks.** We freeze the encoder, and train a linear layer on top to probe the quality of representations. We test on ImageNet classification, as well as on a set of robustness benchmarks.

| Method | Soft loss | Uses labels | ImageNet | ImageNet Real | Background Decomp. | | | MIT States | |
|---|---|---|---|---|---|---|---|---|---|
| | | | | | Same Class | Mixed | ObjectNet | Objects | Attributes |
| SCE | ✓ | ✗ | 71.3 | 78.7 | 61.7 | 58.4 | 20.2 | 44.5 | 31.0 |
| ReSSL (1 crop) | ✓ | ✗ | 69.4 | 76.9 | 56.3 | 53.2 | 18.3 | 44.5 | 31.2 |
| VICReg | ✗ | ✗ | 72.4 | 79.0 | 60.8 | 56.8 | 20.5 | 43.5 | 26.9 |
| Barlow Twins | ✗ | ✗ | 72.8 | 80.0 | 62.7 | 59.4 | 21.6 | **45.9** | **31.7** |
| SimCLR | ✗ | ✗ | 63.4 | 67.8 | 44.7 | 38.9 | 12.1 | 40.9 | 29.1 |
| SupCon | ✗ | ✓ | 74.3 | 79.7 | 64.0 | 59.1 | 24.4 | 45.6 | 30.8 |
| $\mathbb{X}$-CLR | ✓ | ✓ | **75.6** | **81.5** | **66.6** | **62.7** | **27.5** | **45.9** | 31.1 |

ImageNet, we follow SupCon (Khosla et al., 2020) and use AutoAugment (Cubuk et al., 2018). All experiments on the ImageNet dataset were run for 100 epochs with 1024 batch size. The learning rate was set to 0.075 for ImageNet models. For experiments on CC3M and CC12M, we used the standard SimCLR augmentations, and a learning rate of 0.1. The rest of the settings were kept the same. Although SimCLR and SupCon both benefit from longer training, we haven't experimented with training for more epochs due to computational constraints. For more details, see appendix A.7.

In all our experiments, to isolate the effect of our learning objective, we fix the backbone architecture to be a ResNet-50 (He et al., 2015) model as this is the most widely studied, with optimized hyperparameters, for standard contrastive self-supervised learning (Chen et al., 2020). We use the same architecture for CLIP's vision encoder and take advantage of already optimized publicly available checkpoints provided by OpenCLIP (Ilharco et al., 2021) for CC12M. Since no comparable public checkpoint is available for CC3M, we train our own model, see appendix A.6.

We evaluate all models across a suite of benchmarks to gauge how well representations generalize in terms of classification performance. We test on ImageNet classification with standard as well as with ImageNet Real labels, on ImageNet-9 to test robustness to background change (we refer to this as 'Background Decomposition' in our results), on ObjectNet to test robustness to context and view change, and on MIT-States objects and attributes classification to test how well model captures object states. We use linear probing on top of frozen representations for all evaluations, including CLIP. For more details and dataset examples see appendix A.8.

## 4.2 $\mathbb{X}$-SAMPLE CONTRASTIVE WITH WELL-LABELED SAMPLES

We first experiment with $\mathbb{X}$-Sample Contrastive using the well-labeled ImageNet dataset to understand the effect of incorporating similarities across samples in the training objective. We compare $\mathbb{X}$-Sample Contrastive ($\mathbb{X}$-CLR) to a range of baselines: SupCon (Khosla et al., 2020) which uses labels; SCE (Denize et al., 2023) and ReSSL (Zheng et al., 2021) which use self-distillation with soft targets; and to SimCLR (Chen et al., 2020), VICReg (Bardes et al., 2021) and BarlowTwins (Zbontar et al., 2021), which are SSL methods for learning from images.

We find in table 1 representations learned via $\mathbb{X}$-CLR with Sentence Transformer similarities (Reimers and Gurevych, 2019) improve on standard classification performance, with gains of 12.2% relative to SimCLR and 1.3% relative to Supervised Contrastive on ImageNet. We find similar gains when evaluated on revised labels from ImageNet Real of 13.7% and 1.8%, respectively. Finally, we find by capturing similarities across samples, representations learned via $\mathbb{X}$-CLR are more capable of disambiguating objects from backgrounds and attributes with gains on ImageNet-9 (Xiao et al., 2020) and ObjectNet (Barbu et al., 2019). To confirm statistical significance of the results, we report standard deviations over 5 seeds for the models we trained in table 7 (SimCLR, SupCon, and $\mathbb{X}$-CLR). For the remaining ImageNet models, we took published pre-trained encoders, and are limited to one seed.

**Effect of the similarity graph.** We investigate how the choice of similarity graph affects the performance. To calculate the similarity between samples, we use similarity of the captions embedding when using various text encoders: CLIP (Radford et al., 2021), LLama2 (Touvron et al., 2023), Sentence Transformer (Reimers and Gurevych, 2019). We also experiment with the similarity defined by the distance in WordNet hierarchy (Fellbaum, 1998). Baselines include SimCLR Chen et al. (2020), where the similarity graph is the binary augmentation graph, and Supervised Contrastive (Khosla

Table 2: **The effect of the similarity source on the model performance.** We train $\mathbb{X}$-CLR with similarities coming from a variety of sources, and evaluate the representations with linear probing.

| Similarity source | ImageNet | ImageNet Real | Background Decomposition | | ObjectNet |
| --- | --- | --- | --- | --- | --- |
| | | | Same Class | Mixed | |
| Augmentation graph (SimCLR) | 63.4 | 67.8 | 44.7 | 38.9 | 12.1 |
| True class graph (SupCon) | 74.3 | 79.7 | 64.0 | 59.1 | 24.4 |
| Sentence Transformer ($\mathbb{X}$-CLR) | **75.6** | **81.5** | 66.6 | 62.7 | **27.5** |
| CLIP text encoder | 74.4 | 80.6 | **67.5** | **64.2** | 24.5 |
| LLama2 text encoder | 40.9 | 45.8 | 38.3 | 36.0 | 4.3 |
| Distance in WordNet hierarchy | 68.3 | 74.9 | 55.7 | 52.1 | 21.2 |

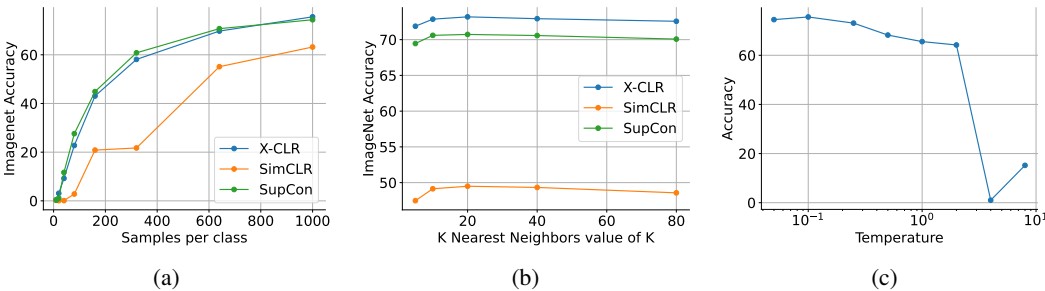

(a)  (b)  (c)

Figure 3: **(a) $\mathbb{X}$-Sample Contrastive learning is data efficient with ImageNet pretraining.** We outperform SimCLR in low data regimes and match Supervised Contrastive trained on ground truth labels at varying levels of data scarcity. **(b) KNN performance ImageNet.** $\mathbb{X}$-CLR outperforms other methods with KNN probing for a range of values of K. **(c) Sensitivity of $\mathbb{X}$-Sample Contrastive to temperature.** We benchmark our method trained with different values of temperature $\tau_s$ on ImageNet.

et al., 2020), where the similarity graph has connections only between samples of the same class. We emphasize that $\mathbb{X}$-CLR is not a model distillation method (Wu et al., 2023) as the *similarities can come from any source*, including any metadata, e.g., WordNet hierarchy in our case or taxonomy of organisms for biological data. For more details regarding this experiment, see appendix A.1. We show results in table 2. Overall, we find that using Sentence Transformer is the best option for classification performance, and use this similarity in all the following experiments.

**Can we improve contrastive learning under data scarcity?** To answer this question, we train all three models SimCLR, SupCon, and $\mathbb{X}$-CLR by varying the number of samples seen for each class in ImageNet. We find $\mathbb{X}$-CLR, by incorporating class labels and how they relate, is able to learn representations with comparable performance to SupCon trained with ground truth class labels and outperform SimCLR even when few training samples are available per class as shown in fig. 3a.

### 4.3 $\mathbb{X}$-SAMPLE CONTRASTIVE LEARNING WITH NOISY MULTIMODAL SAMPLES

Contrastive loss also plays a pivotal role in multimodal vision-language models such as CLIP. The contrastive training objective matches noisy caption-image pairs. Here we experiment with $\mathbb{X}$-Sample Contrastive loss by using the noisy captions to learn similarities across samples. We compare both SimCLR as a standard contrastive model and CLIP trained on the same caption-image data across two levels of scale: 3 and 12 million samples from CC3M and CC12M.

We find incorporating $\mathbb{X}$-CLR loss leads to representations with higher classification accuracy and disambiguation of objects from their attributes and backgrounds. With CC12M training shown in table 3, $\mathbb{X}$-Sample Contrastive learning outperforms SimCLR by 0.5% and CLIP by 0.6% with CC12M with similar gains for ImageNet Real. We also find $\mathbb{X}$-CLR training can better disambiguate object foreground from backgrounds, with gains of 0.6-1.5% over SimCLR and 3.3-5.6% over CLIP.

We find learning similarites across samples with $\mathbb{X}$-CLR leads to more considerable gains when less data is available. When trained on CC3M, $\mathbb{X}$-CLR outperforms SimCLR by 1.2% and CLIP by 17.2% on ImageNet, with similar gains on ImageNet Real as shown in table 3. We find $\mathbb{X}$-CLR training can

Table 3: $\mathbb{X}$-**Sample Contrastive learning performance with CC3M and CC12M training.** We train $\mathbb{X}$-CLR on conceptual caption images with text caption similarities, and compare to CLIP and SimCLR. SimCLR does not use text data and is only trained on images.

| Data | Method | ImageNet | ImageNet Real | Background Decomposition | | |
| | | | | Same Class | Mixed | ObjectNet |
|---|---|---|---|---|---|---|
| | SimCLR | 57.0 | 64.0 | 24.4 | 18.9 | 10.8 |
| CC3M | CLIP | 41.0 | 47.6 | 12.5 | 10.6 | 7.8 |
| | $\mathbb{X}$-CLR | **58.2** | **65.6** | **26.7** | **20.3** | **11.5** |
| | SimCLR | 58.9 | 66 | 24.6 | 19.8 | 12.7 |
| CC12M | CLIP | 58.8 | 66.1 | 20.5 | 17.1 | 11.9 |
| | $\mathbb{X}$-CLR | **59.4** | **66.7** | **26.1** | **20.4** | **13.4** |

Table 4: $\mathbb{X}$-**CLR performance when used to finetune pretrained models.** We take a SimCLR checkpoint and fine-tune it using $\mathbb{X}$-CLR objective for 10 epochs. We then evaluate the original checkpoint and the fine-tuned one using linear probing.

| | ImageNet | ImageNet Real | Background Decomposition | | |
| | | | Same Class | Mixed | ObjectNet |
|---|---|---|---|---|---|
| SimCLR | 63.4 | 67.8 | 44.7 | 38.9 | 12.1 |
| + $\mathbb{X}$-CLR finetuning | **66.5** | **74.4** | **53.9** | **50.0** | **17.4** |

more considerably disambiguate object foregrounds from backgrounds compared to CLIP when less training data is available, with gains of 10.3-14.2% over CLIP.

## 4.4 $\mathbb{X}$-SAMPLE CONTRASTIVE CAN BE USED TO FINETUNE PRETRAINED BACKBONES

We validate whether $\mathbb{X}$-CLR can be used as a finetuning objective for pretrained backbones, given the growing abundance of publicly available model checkpoints. Here, we evaluate a pretrained SimCLR model by finetuning for 10 epochs on ImageNet with $\mathbb{X}$-CLR instead of the original SimCLR contrastive objective. We see in table 4 finetuning with $\mathbb{X}$-CLR improves classification performance on ImageNet by 3.1% and on ImageNet Real by 6.6%. Furthermore, we see by relating samples during the finetuning stage, $\mathbb{X}$-CLR can disambiguate object foregrounds from backgrounds with grains of 9.2-11.1% on ImageNet-9 as well as improvements on natural object transformations from ObjectNet with a gain of 5.3% after finetuning.

## 4.5 $\mathbb{X}$-SAMPLE CONTRASTIVE OBJECTIVE INTRODUCES ONLY MINIMAL COMPUTATIONAL OVERHEAD

Both for ImageNet and conceptual captions datasets, we don't run the text encoder for each sample we see, and instead precompute the similarity values. For more details, see appendix A.7. Avoiding running the text encoder during model training avoids the extra overhead at the price of some pre-processing. Pre-processing takes less than 2 hours for CC12M when using one GPU, about 30 minutes for CC3M, and less than 5 minutes for ImageNet. To further analyze how much overhead there is, we compare the average time it takes to process one batch for SimCLR and $\mathbb{X}$-CLR. The results are shown in table 5. Overall, we didn't notice any significant difference in the amount of time it takes to train models with the $\mathbb{X}$-CLR objective compared to the regular contrastive objective. To train on ImageNet, we used 8 Nvidia V100s, and each run took about 30 hours. With the same setup, CC3M runs took about 50 hours, and CC12M runs took roughly 9 days.

## 5 ANALYZING REPRESENTATIONS LEARNED WITH $\mathbb{X}$-CLR

**KNN Clustering.** To confirm the representations learned via $\mathbb{X}$-CLR also work well for downstream tasks with non-linear decision boundaries, we perform evaluation using the common K-nearest neighbor (KNN) protocol. The results shown in fig. 3b demonstrate $\mathbb{X}$-CLR outperforms both

Table 5: **Analyzing the computation overhead of the $\mathbb{X}$-Sample Contrastive objective during training.** We measure the mean and standard deviation of the time needed to process one batch update for $\mathbb{X}$-CLR and SimCLR. $\mathbb{X}$-CLR introduces nearly no computational overhead.

| Method | Seconds per batch ImageNet | Seconds per batch CC |
|---|---|---|
| SimCLR | $0.866 \pm 0.008$ | $0.874 \pm 0.034$ |
| $\mathbb{X}$-CLR | $0.866 \pm 0.010$ | $0.877 \pm 0.032$ |

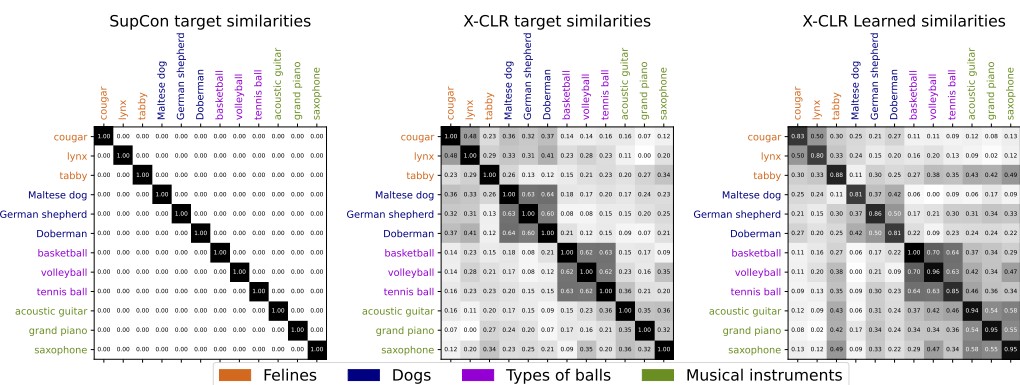

Figure 4: **Visualizing pairwise similarities** SupCon (Khosla et al., 2020) objective does not encourage non-zero similarity between samples of different classes (left), while $\mathbb{X}$-CLR target similarities take into account semantic closeness within categories such as dogs or types of balls (center). On the right, we see that the trained model successfully learns the soft similarity. For more graphs, see fig. 6.

SimCLR and SupCon baselines across a range of choices for $K$. We also show KNN results for models trained on conceptual captions in appendix A.4.

**Analyzing the learned graph from $\mathbb{X}$-Sample Contrastive representations.** Here we examine whether the learned representations from $\mathbb{X}$-Sample Contrastive capture semantically meaningful similarities. To do so, we select four groups of three ImageNet classes: felines, dogs, types of balls, and musical instruments. For each pair of classes, we then compare the representation similarities using cosine similarity. A higher average pairwise similarity indicates the model's latent representations encode the classes similarly. In fig. 4 we show the graph of similarities learned after training with $\mathbb{X}$-CLR on ImageNet. We find that the image encoder successfully captures the similarity within the class groups. Additionally, we repeat the analysis of the learned similarity graph in (Zhang et al., 2023). In that work, the authors also view contrastive learning through the lens of graphs, and introduce metrics to analyze the quality of the learned graph. In appendix A.3, table 9, we find that $\mathbb{X}$-CLR learns a better structured graph compared to CLIP, SimCLR, and SupCon, with samples better connected within the class, and less connected for unrelated classes. We also plot T-SNE (Van der Maaten and Hinton, 2008) projections of $\mathbb{X}$-CLR, SimCLR, and SupCon representations trained on ImageNet in fig. 13 and find that $\mathbb{X}$-CLR learns well-structured embeddings.

**The effect of softmax temperature, and inferred similarity graph.** We show the sensitivity of $\mathbb{X}$-CLR to temperature $\tau_s$ in fig. 3c on ImageNet. In the limit, when temperature goes to 0, we recover Supervised Contrastive method for ImageNet, or SimCLR in case of conceptual captions, see appendix A.9 for a more detailed explanation. With low temperature, the similarity is 1 only if the captions are exactly the same. As the temperature increases, more weight is put on the soft positives compared to the true positives (i.e. augmentations of the same sample). With high temperature, our method is unstable as too much emphasis is put on the soft positive examples compared to the true positives. We find that the value of 0.1 strikes the optimal balance and provides an improvement over pure Supervised Contrastive objective, while still emphasizing true positives enough. We show how $\tau_s$ changes the objective in fig. 5b. We see that more probability mass is put on the true positive for low temperatures, with ImageNet on average having lower true positive weight than CC3M samples. In fig. 5a, we see that on average, similarity between samples in ImageNet is much higher than in CC3M, which may be caused by inadequate caption quality in CC3M.

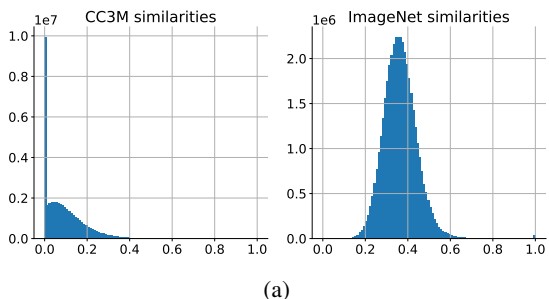 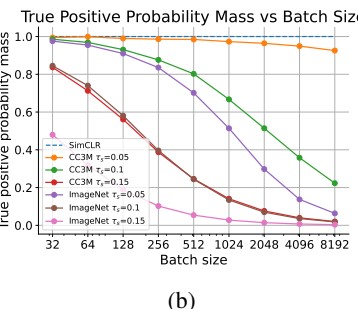

(a)              (b)

Figure 5: (a) Histograms of the similarities calculated using Sentence Transformer on ImageNet and CC3M. While for ImageNet the average similarity is around 0.35, it is much lower on CC3M, signifying that the graph contains less information for CC3M. (b) Effect of the temperature and batch size on the weight assigned to the true positvie.

Table 6: **The effect of label quality matters for fine-grained attribute disambiguation.** We evaluate $\mathbb{X}$-CLR trained on high-quality ImageNet labels and on noisier CC labels, and find that smaller but better quality ImageNet dataset yields better representations for attribute disambiguation on MIT states. For more details on the benchmark, see A.8.3.

| Pretraining | Data Size | Quality | MIT States Attributes | MIT States Objects |
|---|---|---|---|---|
| CLIP CC3M | 3M | Noisy | 27.0 | 40.1 |
| CLIP CC12M | 12M | Noisy | 23.3 | 36.9 |
| $\mathbb{X}$-CLR CC3M | 3M | Noisy | 29.5 | 40.7 |
| $\mathbb{X}$-CLR CC12M | 12M | Noisy | 30.1 | 42.1 |
| $\mathbb{X}$-CLR ImageNet | 1M | High | **30.9** | **45.8** |

**The impact of label quality for fine-grained attribute disambiguation** We show in table 6 how label quality can impact downstream performance on finer-grained attribute disambiguation. We find labels from noisy captions degrade performance for fine-grained object attributes in MIT States (Isola et al., 2015) for both $\mathbb{X}$-CLR and CLIP. We find $\mathbb{X}$-CLR with high quality labels from ImageNet, can outperform models trained on much larger noisier data. Compared to CLIP trained on $12\times$ more data, $\mathbb{X}$-CLR achieves 30.9% vs. 23.3% for CLIP on attribute classification and 45.8% vs. 36.9% for CLIP on object classification on MIT States benchmark (see appendix A.7 for more details).

## 6 DISCUSSION

We revisited the graph view on the commonly used contrastive learning methods and developed a better learning objective, $\mathbb{X}$-CLR, by integrating a soft similarity graph. The adjacency matrix of the proposed graph contains not just 0 and 1, but also any values between, capturing the degree of similarity *across* samples. We experiment with different ways of constructing the graph, and find that indeed we can build a soft graph that improves over the existing binary graph of contrastive methods. However, we believe that there are better ways of constructing the graph than what we found, particularly for the conceptual captions dataset where the captions are quite noisy. A better graph can possibly be built using other metadata, such as location or time. We also believe that ideas from $\mathbb{X}$-CLR can be used with other modalities where extra metadata is available, e.g. biological data with the associated taxonomy. The soft graph can also be used to enhance non-contrastive objectives such as BYOL (Grill et al., 2020) or VICReg (Bardes et al., 2021).

**Limitations.** The main limitation of the present work is that constructing the cross-sample similarity graph requires extra data, as well as some extra memory to store it. When the extra data is not available, the only options remaining are to build the graph using the augmentations, self-distillation, or other pre-trained models. The resulting method is also highly dependent on the quality of the graph, as we have seen with conceptual captions datasets.

## 7    ACKNOWLEDGMENTS

This material is based upon work supported by the National Science Foundation under NSF Award 1922658.

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

Table 7: **Analyzing statistical significance of ImageNet results.** Each experiment is ran with 5 seeds, we report the mean and standard deviation.

| Method | ImageNet | ImageNet Real | Background Decomposition | | | MIT States | |
| | | | Same Class | Mixed | ObjectNet | Objects | Attributes |
|---|---|---|---|---|---|---|---|
| SimCLR | $63.43 \pm 0.12$ | $67.75 \pm 0.27$ | $38.88 \pm 0.43$ | $44.67 \pm 0.60$ | $12.07 \pm 0.33$ | $40.92 \pm 0.26$ | $29.08 \pm 0.17$ |
| SupCon | $74.30 \pm 0.16$ | $79.66 \pm 0.12$ | $59.08 \pm 0.44$ | $64.00 \pm 0.62$ | $24.42 \pm 0.25$ | $45.56 \pm 0.16$ | $30.83 \pm 0.20$ |
| $\mathbb{X}$-CLR | $75.56 \pm 0.09$ | $81.54 \pm 0.13$ | $62.74 \pm 0.27$ | $66.59 \pm 0.25$ | $27.53 \pm 0.13$ | $45.86 \pm 0.15$ | $31.10 \pm 0.18$ |

## A  APPENDIX / SUPPLEMENTAL MATERIAL

### A.1  MORE LEARNED SIMILARITIES COMPARISONS

We experiment with building the graph in the following ways:

- Graph with connections only between samples of the same class (SupCon);
- Graph with connections only between augmentations of the same image (SimCLR);
- Graph where soft similarity is inferred by comparing representations of the sample captions. The representations are computed using the sentence transformer (Reimers and Gurevych, 2019), CLIP text encoder (Radford et al., 2021), LLama2 encoder (Touvron et al., 2023). For LLama2, we averaged the output tokens;
- Graph where the connection strength is defined by the distance in WordNet (Fellbaum, 1998) hierarchy. We used the NTLK library Bird et al. (2009), and used the Wu-Palmer similarity (Wu and Palmer, 1994) between the class synsets;
- We also experiment with random graphs where the cross-sample connections' strengths are fully random. The connections are either random per sample pair, or random per class pair (akin to using a random text encoder for captions).

The results are shown in table 8. We find that overall, the Sentence Transformer graph performs the best, although the CLIP text encoder achieves good performance as well. Interestingly, we find that using WordNet hierarchy distance did not work well. Additionally, random per class pair performs quite well, particularly on background decomposition. We hypothesize that it's due to a regularizing effect of the random similarities. Nevertheless, the performance of ImageNet, ImageNet Real and Object Net is lower than when using Sentence Transformer. We visualize learned and target similarities for SupCon graph and for the graph built using CLIP text encoder in fig. 6.

**Visualising similarities**  In fig. 4, to visualize learned similarities, for each class we pick 100 examples from the dataset and encode them. Then, to calculate the average learned similarity between two classes, we take the 100 examples for each of the two classes, and calculate the Cartesian product, yielding 10,000 similarities. We take the mean over those 10,000 similarities to represent the average learned similarity for a class pair.

**Similarities when training on CC datasets**  In appendix A.2, we show the similarities learned by $\mathbb{X}$-CLR on CC3M and CC12M datasets.

### A.2  ANALYZING STATISTICAL SIGNIFICANCE OF THE RESULTS

To make sure the difference in performance we observe is statistically significant, we run $\mathbb{X}$-CLR, SimCLR, and SupCon pretraining with 5 different seeds. We report the results of the evaluations in table 7.

### A.3  ANALYZING THE LEARNED GRAPH

We follow the analysis of Zhang et al. (2023) and show results in table 9. The analysis studies two values: label error which measures how similar samples of different classes are on average, and intra-class connectivity, which measures the similarity of the samples within the class relative to those

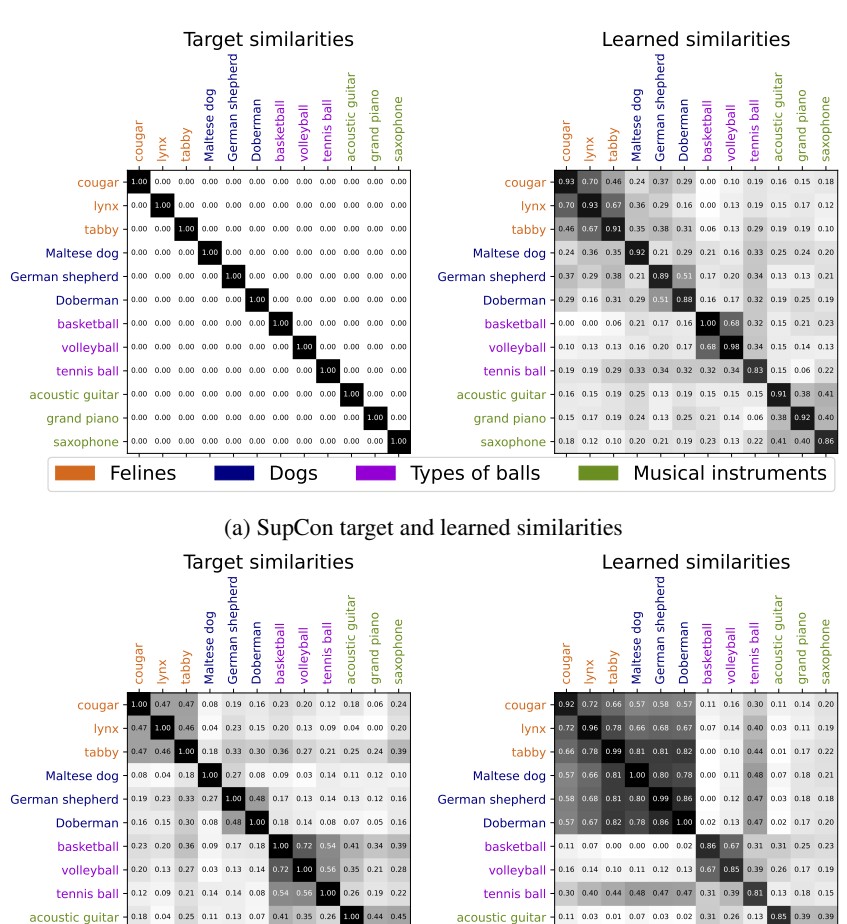

(a) SupCon target and learned similarities

(b) CLIP target and learned similarities

Figure 6: Target and learned similarities for different graphs.

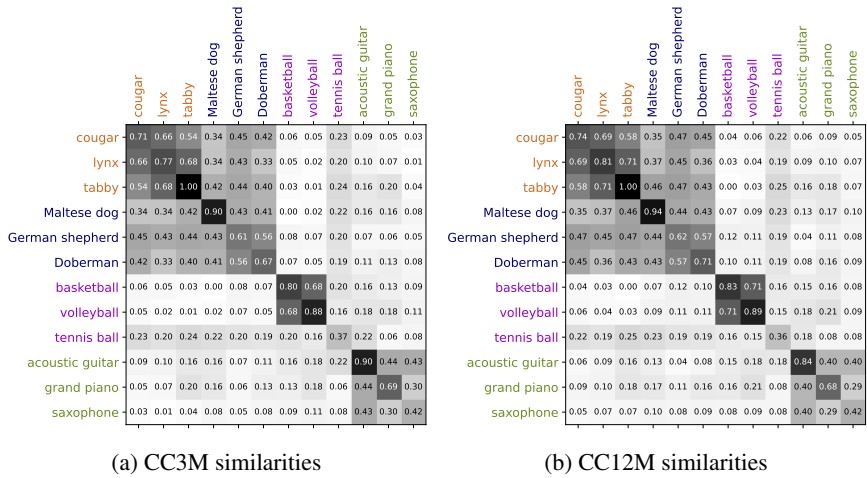

(a) CC3M similarities

(b) CC12M similarities

Figure 7: $\mathbb{X}$-CLR Learned similarities when trained on **a)** CC3M and **b)** CC12M.

Table 8: The effect of the similarity source on the model performance.

| Similarity source | ImageNet | ImageNet Real | Background Decomposition | | ObjectNet |
|---|---|---|---|---|---|
| | | | Same Class | Mixed | |
| Augmentation graph (SimCLR) | 63.4 | 67.8 | 44.7 | 38.9 | 12.1 |
| True class graph (SupCon) | 74.3 | 79.7 | 64.0 | 59.1 | 24.4 |
| Sentence Transformer ($\mathbb{X}$-CLR) | **75.6** | **81.5** | 66.6 | 62.7 | **27.5** |
| CLIP text encoder | 74.4 | 80.6 | 67.5 | 64.2 | 24.5 |
| LLama2 text encoder | 40.9 | 45.8 | 38.3 | 36.0 | 4.3 |
| Random per class pair | 74.5 | 80.8 | **71.0** | **68.0** | 26.6 |
| Random per sample pair | 0.1 | 0.1 | 0 | 0 | 0 |
| Distance in WordNet hierarchy | 68.3 | 74.9 | 55.7 | 52.1 | 21.2 |

Table 9: Analyzing the learned representations' connectivity

| Metric | CLIP | SimCLR | SupCon | X-CLR |
|---|---|---|---|---|
| Label error ($\downarrow$) | 0.550 | 0.250 | 0.250 | 0.223 |
| Intra-class connectivity ($\uparrow$) | 1.233 | 1.700 | 2.005 | 2.193 |

from different classes. This allows us to determine how well the learned graph captures the class relationships in the data. Since the open-source repository of that paper did not contain the code for analysis, we re-implemented it to the best of our ability.

Here, $\mathbb{X}$-CLR, SimCLR, and SupCon are trained on ImageNet, while CLIP is trained on CC12M. According to these metrics, $\mathbb{X}$-CLR representation is the best among the baselines, meaning that we learn a better structured graph. We note that our SimCLR numbers are much better than in the original paper. We suspect that it's due to the fact that the authors train SimCLR with the batch size of 512, while we use 2048. ImageNet classification performance of our SimCLR model is also higher, at 63.4, compared to 61.2.

We also note that label error, which is the measure of average similarity between instances of different classes, is lower for our method, although the loss itself encourages it to be higher for related samples. This is due to the fact that in this analysis, we use the first 10 classes from ImageNet (replicating the original procedure), and those classes are not related to each other.

## A.4 KNN EVALUATION

Apart from testing the models trained on ImageNet using KNN, we also evaluate the models trained on CC3M and CC12M. The results are shown in fig. 8. We see that $\mathbb{X}$-CLR performs better on CC3M, and comparatively with SimCLR when trained on CC12M.

## A.5 IMAGENET-9 DETAILS

## A.6 CLIP DETAILS

In CC3M experiments, we train the model from scratch, as OpenCLIP didn't have a checkpoint trained on that dataset. We trained both for 32 and 100 epochs, and found that the model trained for 100 epochs performs better. Since 32 epochs is the default CLIP number of epochs, we also report results for 32 epochs. The results are shown in table 10.

## A.7 MORE TRAINING DETAILS

We train SimCLR, SupCon and $\mathbb{X}$-CLR using the LARS optimizer (You et al., 2017). In all cases, we use the same ResNet-50, with a two layer projector on top. The output dimension of the projector is 128. We note that SupCon benefits from bigger batch sizes and longer training as was shown in (Khosla et al., 2020), but in our experiments we did not explore training for more than 100 epochs or

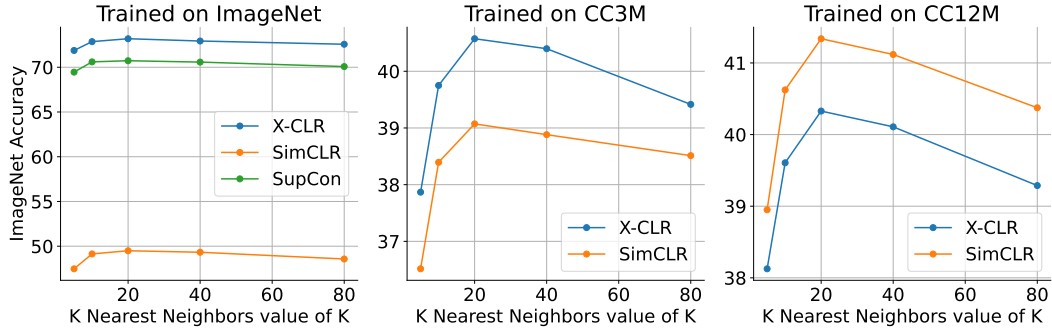

Figure 8: Results of models trained on ImageNet, CC3M, CC12M on ImageNet validation when using KNN classifier.

Table 10: **CLIP on CC3M** We train our own models on CC3M and find that training longer improves the performance. Nevertheless, CLIP struggles with small datasets.

| Method | ImageNet | ImageNet Real | Background Decomposition | | ObjectNet |
| --- | --- | --- | --- | --- | --- |
| | | | Same Class | Mixed | |
| CLIP 100 epochs | 41.0 | 47.6 | 12.5 | 10.6 | 7.8 |
| CLIP 32 epochs | 36.8 | 42.0 | 11.5 | 9.8 | 6.0 |

with bigger batch sizes. For fair comparison, we keep the number of epochs to 100 and batch size to 1024 across all methods we train on ImageNet.

**Fetching similarities** For ImageNet, since the number of classes is known, we pre-compute the similarity matrix of dimension $1000 \times 1000$, and retrieve elements from it depending on the associated class labels for a given sample pair to obtain the similarity value. For conceptual captions, we run the text encoder on the full dataset and save the encodings to disk. Then, when loading an image from disk, we also load the associated encoding of the corresponding caption. The similarity matrix for a given batch is then obtained by calculating the Cartesian product of those encodings.

**MIT States** In order to evaluate on this dataset using linear probing, we split the dataset randomly into two even parts, one used for training the linear layer, the other for evaluation. We train separately to classify objects and attributes.

## A.8 EVALUATION DETAILS

All evaluations are done with linear probing. We use the learning rate of 1 and plain SGD optimizer for all evaluations. We do not use zero-shot classification for CLIP, or any other models. Since $\mathbb{X}$-CLR doesn't align the representations of text and images, we cannot utilize zero-shot classification to evaluate. For ImageNet models we train ourselves, we do not train the linear prober separately, we train it along with the rest of the model with detached gradients.

### A.8.1 IMAGENET REAL

ImageNet Real evaluation uses improved ImageNet labels from (Beyer et al., 2020). The improved evaluation allows for multiple labels for scenes where labels are ambiguous, and corrects some mistakes in annotations.

### A.8.2 BACKGROUND DECOMPOSITION WITH IMAGENET-9

ImageNet-9 (Xiao et al., 2020) proposes multiple benchmarks to test model robustness to the background perturbation. The benchmark is created by taking samples from ImageNet, segmenting

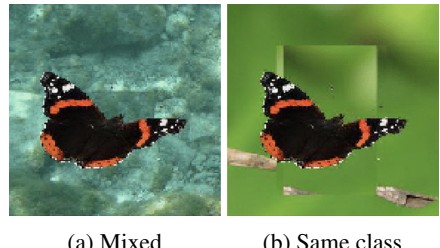

(a) Mixed      (b) Same class

Figure 9: The two settings we use for ImageNet-9 (Xiao et al., 2020) benchmark. In mixed setting in a), the background is swapped with that of a sample belonging to another class. In a), this results in an image of a butterfly on underwater background. Performing well in this setting is hard if the model learns spurious correlations with the background. In same class setting in b), the background is swapped with that of a sample belonging to the same class. In b), we see a butterfly pasted onto the background of another butterfly sample, in this case a tree branch on green background. This lets us see how much swapping the background affects the performance and serves as a baseline for 'mixed'.

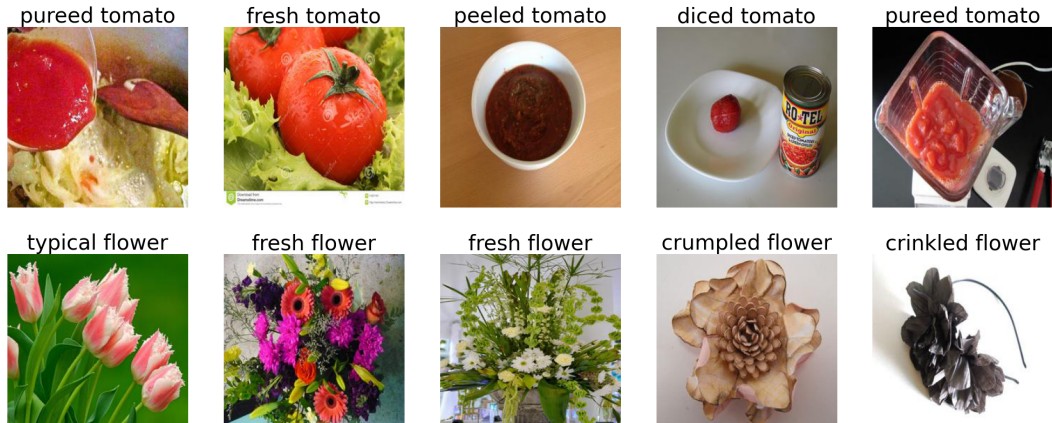

Figure 10: MIT States (Isola et al., 2015) samples from two classes: tomato and flower. Each sample has an associated class name and attribute. We show tomatoes and flowers classes with different attributes. For example, the attributes we show for tomato are "pureed", "fresh", "peeled", and "diced". We train our models to classify class names and attributes. When predicting attributes, we do not condition on class names and vice versa.

the object in the scene, and swapping out the background. Since the benchmark uses the same classes as ImageNet, we do not retrain the ImageNet classifier. We consider two setups for evaluation:

- where the object is seen with backgrounds from the same class, which we label "same class";

- where the objects are seen with backgrounds from other classes, which we label "mixed".

These evaluations isolate how well the model can classify an object without relying on spurious associations from the background. We show an example in fig. 9.

### A.8.3 MIT STATES

Here we measure object classification accuracy as the object attributes vary as well as the models' ability to identify the object attributes such as color, texture, shape. We show a few examples from the dataset in fig. 10. Because this benchmark uses a different set of classes from ImageNet, we need to train a classifier separately.

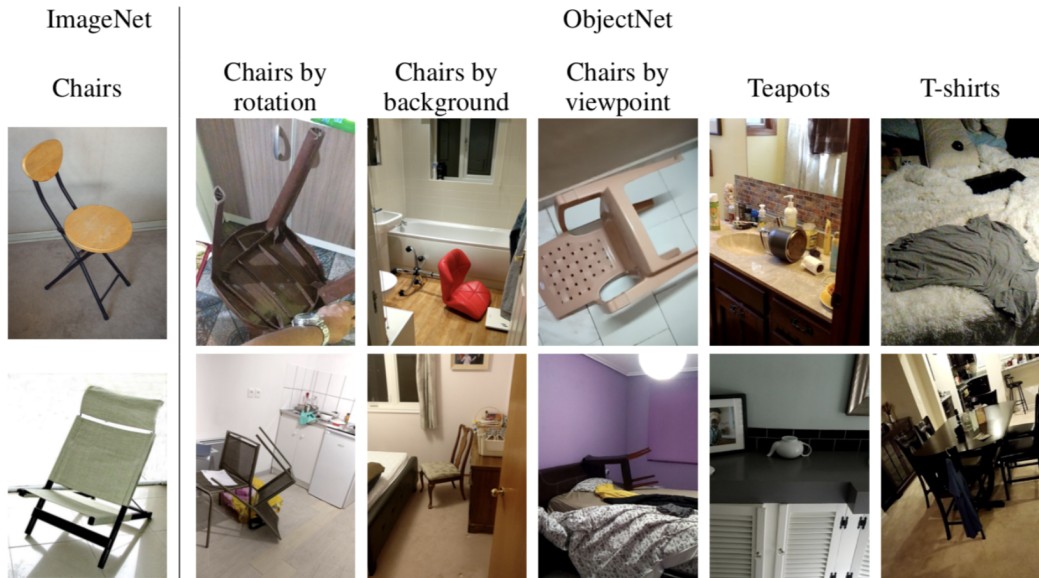

Figure 11: Examples from ObjectNet dataset. We take the figure from Barbu et al. (2019). As opposed to ImageNet samples, ObjectNet shows objects in a unexpected contexts and positions. For example, a chair lying on the bathroom floor or a teapot on its side near a bathroom sink. This benchmark tests robustness of the models to these unexpected context changes.

### A.8.4 OBJECTNET

ObjectNet contains real-world objects with varying poses, backgrounds, and viewpoints. The benchmark uses the same classes as ImageNet, and measures how robust the model is to seeing the objects in unexpected contexts. Like for ImageNet-9, since the benchmark uses the same classes as ImageNet, we do not retrain the ImageNet classifier. We show examples in fig. 11.

### A.9 CONNECTION BETWEEN SUPERVISED CONTRASTIVE LEARNING AND $\mathbb{X}$-CLR

Here, we will outline how as the temperature $\tau_s$ approaches 0, $\mathbb{X}$-CLR becomes SupCon. Supervised Contrastive Learning (Khosla et al., 2020) also uses image augmentations, and augments each image twice, to obtain what they call "a multiviewed batch". Then, in equation 2, they propose the loss:

$$\mathcal{L}_{\text{out}}^{\text{sup}} = \sum_{i \in I} \mathcal{L}_{\text{out},i}^{\text{sup}} = \sum_{i \in I} \frac{-1}{|P(i)|} \sum_{p \in P(i)} \log p_{i,p}$$

where $p_{i,j}$ is defined as follows:

$$p_{i,j} = \frac{\exp(\text{sim}(z_i, z_j)/\tau)}{\sum_{k=1}^{2N_b} \mathbf{1}_{[k \neq i]} \exp(\text{sim}(z_i, z_k)/\tau)}$$

However, $|P(i)|$ is exactly the number of positive samples, and $p_{i,p}$ is the probability of $i$ and $p$ being a positive pair according to the model. We set $s_i^{\text{supcon}}$ to be a distribution over $2N_b - 1$ candidates for positive pairs and define it as follows:

$$s_{i,j}^{\text{supcon}} = \begin{cases} \frac{1}{|P(i)|}, & \text{if } j \in P(i) \\ 0, & \text{otherwise} \end{cases}$$

Then, we can write down the original loss as:

$$\mathcal{L}_{\text{out},i}^{\text{sup}} = H(s_i^{\text{supcon}}, p_i)$$

where H is the cross-entropy. This looks exactly like the $\mathbb{X}$-CLR objective. We can recover SupCon objective if we increase the temperature $\tau_s$: the resulting distribution $s_i$ will be equal to $s_i^{\text{supcon}}$.

```
def training_step(images, sims):
    # encode images with the model we are training
    img_emb = img_enc(images)  # N x D_img
    # convert targets to distributions
    tgt_sims = softmax(sims / tao_s, dim=1)
    # calculate similarities
    img_sims =
        softmax(img_emb @ img_emb.T / tao, dim=1)
    # the loss is the cross-entropy
    return CE(tgt_sims, img_sims, dim=1).mean()
```

Figure 12: Pseudocode for $\mathbb{X}$-CLR loss for general similarities, not necessarily coming from a text encoding similarities.

## A.10 CONNECTION BETWEEN CLIP AND $\mathbb{X}$-CLR

$\mathbb{X}$-CLR and CLIP are similar because they both use images and text data. That said, $\mathbb{X}$-CLR trains only the image representation, and the text is used only to calculate target similarities. The similarities do not have to come from text, as we showed in table 2. CLIP, on the other hand, trains both the image and text encoders. CLIP loss can be expressed as follows:

$$\mathcal{L}_{\text{CLIP}} = \mathcal{L}_{\text{text}} + \mathcal{L}_{\text{image}}$$

$$p_{i,j}^{\text{text}-\text{image}} = \frac{\exp(\text{sim}(z_i^{\text{text}}, z_j^{\text{image}})/\tau)}{\sum_{k=1}^{N_b} \exp(\text{sim}(z_i^{\text{text}}, z_k^{\text{image}})/\tau)}$$

$$\mathcal{L}_{\text{text}} = \frac{1}{N_b} \sum_{i=1}^{2N_b} H(\mathbb{1}_i, p_i)$$

$$p_{i,j}^{\text{image}-\text{text}} = \frac{\exp(\text{sim}(z_i^{\text{image}}, z_j^{\text{text}})/\tau)}{\sum_{k=1}^{N_b} \exp(\text{sim}(z_i^{\text{image}}, z_k^{\text{text}})/\tau)}$$

$$\mathcal{L}_{\text{image}} = \frac{1}{N_b} \sum_{i=1}^{2N_b} H(\mathbb{1}_i, p_i^{\text{image}-\text{text}})$$

Here, $N_b$ is the batch size, $D$ is the dimension of the encodings, $\boldsymbol{Z}^{\text{image}} \in \mathbb{R}^{N_b \times D}$ is the batch of image encodings coming from the visual encoder, $\boldsymbol{Z}^{\text{text}} \in \mathbb{R}^{N_b \times D}$ is the batch of text encodings coming from the text encoder model. $\mathbb{1}_i$ is a one-hot distribution on $N_b$ elements, with all the probability on $i$-th element.

We can see that as opposed to $\mathbb{X}$-CLR loss, CLIP loss has two major differences:

- CLIP loss is pushing similarities of images and text encodings together for positive pairs, while $\mathbb{X}$-CLR loss only pushes the images representations to each other. $\mathbb{X}$-CLR only works with image encodings, and beyond the target similarities, doesn't deal with text encodings at all;

- $\mathbb{X}$-CLR loss uses a soft target distribution in cross entropy, while CLIP uses a one-hot distribution as target.

Due to these fundamental differences, $\mathbb{X}$-CLR is different from CLIP even when the target similarities are coming from a frozen pre-trained CLIP text encoder. We hypothesize that CLIP could also benefit from soft target similarities, but leave that extension for future work.

## A.11 PSEUDOCODE FOR THE GENERAL SIMILARITIES

As we showed in table 2, the similarities do not have to come from a text encoder. In fig. 12, we show pseudocode for how the loss function would look like general target similairites provided in variable `sims`.

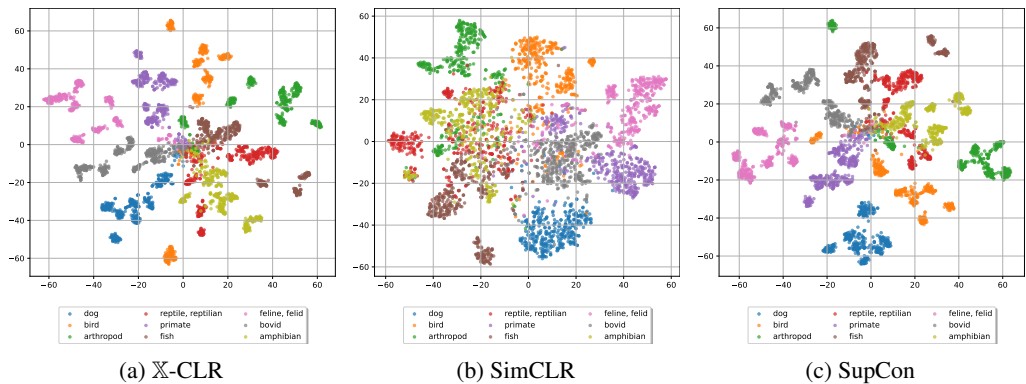

(a) 𝕏-CLR       (b) SimCLR       (c) SupCon

Figure 13: 𝕏-CLR and SupCon representations fall into a well-defined clusters, whereas SimCLR representations are less structured.

## A.12 T-SNE OF THE LEARNED REPRESENTATIONS

In fig. 13, we show T-SNE plots of representations of a few superclasses from ImageNet. We used the 'living 9' set of classes from (Engstrom et al., 2019).

