# OpenReview forum: "$\mathbb{X}$-Sample Contrastive Loss: Improving Contrastive Learning with Sample Similarity Graphs"
_ICLR.cc/2025/Conference — ICLR 2025 Poster_

### Official Review · Reviewer_MXJi · 2024-10-28

**Soundness:** 3
**Presentation:** 3
**Contribution:** 2
**Rating:** 6
**Confidence:** 4

**Summary:**

The paper argue for revising the standard contrastive cross-entropy loss from binary targets of a sparse augmentation graph to a soft target graph where the target values can be between [0, 1]. The new method is called X-CLR. The authors suggest to use extra metadata associated with the dataset to find the soft target distribution, and in this work mostly use the pairwise similarities from a pretrained text encoder on text captions/descriptions to encode this distribution. Although the authors note that the framework is general, "the similarities can come from any source including any metadata". Conduct experiments to train vision models with text captions/descriptions. The study spans different datasets and the results show improvements over contrastive baselines.

**Strengths:**

1. The paper is well-written and easy to read.
2. Experiment study spans different datasets and comparisons to contrastive baselines.
3. The authors test with different similarity graphs (random graph, graph inferred from metadata, augmentation graph, true class graph), and find that a soft graph can improve over the binary graph of existing contrastive approaches.

**Weaknesses:**

1. Relying on extra metadata and a pretrained model limits the applicability compared to the contrastive baselines. I also think this is a simple solution.
2. The idea of soft-targets for cross-entropy based contrastive loss is very general. It also lack some novelty to the paper's related work on soft-targets.

**Questions:**

Suggestion: Finding a soft target graph without extra metadata or pretrained model with self-supervised techniques would give better applicability.

---

> ### Author Response · Authors · 2024-11-19
> **Response**
>
> Thank you for your feedback! Below, we address your questions and concerns. We remain available to answer any additional questions you may have.
>
> > **Relying on extra metadata and a pretrained model limits the applicability compared to the contrastive baselines. I also think this is a simple solution.**
>
> We agree that relying on extra metadata is limiting compared to traditional SSL methods. However, in this paper **we focus on the best way to introduce the similarity information into training** when it's available, and propose an objective for doing so. We find that this helps learn better representations compared to SupCon on ImageNet and CLIP on Conceptual Captions.
>
> > **The idea of soft-targets for cross-entropy based contrastive loss is very general. It also lack some novelty to the paper's related work on soft-targets**
>
> We agree that related work has used soft targets, but that was done mostly for distillation. In our work, we focus on introducing the similarities without emphasis on where they come from. We experiment with different sources of similarities, including WordNet hierarchy distance, which doesn't require running another model at all. This approach, to the best of our knowledge, is novel, and tackles a different problem from distillation methods.
>
> > **Suggestion: Finding a soft target graph without extra metadata or pretrained model with self-supervised techniques would give better applicability.**
>
> Thank you for the suggestion! We agree that methods that do not require metadata are less restricted in their applications, but in this work we explicitly focus on introducing the extra metadata by means of cross-sample similarities. The goal of this work is to show that modeling cross-sample similarities leads to better representations. This of course assumes having a source of those similarities.  We agree that deducing similarities without needing metadata is an interesting direction for future work.

---

> > ### Author Response · Authors · 2024-11-25
> >
> > Dear reviewer MXJi,
> >
> > We would like to remind you that the the end of the discussion period is in less than 2 days. We would appreciate it if you could go over our response when you get a chance. We believe we addressed your questions and concerns, but if you have further questions, we remain available to answer them. We kindly request that you take the response into account when reconsidering the score of the paper.

---

> > > ### Author Response · Authors · 2024-12-02
> > >
> > > Dear reviewer,
> > >
> > > We would like to remind you that the the end of the discussion period is in less than 24 hours. We would really appreciate it if you could go over our response and let us know if it addresses your concerns.

---

> > > > ### Comment · Reviewer_MXJi · 2024-12-02
> > > >
> > > > I thank authors for clarification and responses. I still have concerns with the novelty, highlighted by Reviewer QWUr, that the novelty rather should come from interesting choices for the source of metadata. However I think the paper has aspects that can have some value for contrastive learning in future work. For that reason I will maintain my rating.

---

### Official Review · Reviewer_R31V · 2024-10-30

**Soundness:** 3
**Presentation:** 3
**Contribution:** 4
**Rating:** 8
**Confidence:** 3

**Summary:**

This paper introduces a new objective, called $\mathbb{X}$-Sample Contrastive Learning, where the binary designations in standard contrastive learning are replaced by a soft target based on similarities between samples. These similarities can be obtained through any preferred method, with this paper mainly using frozen text encoders for the experiments. Moreover, the method can be applied fairly easily to any contrastive objective, with this paper focusing on the InfoNCE objective.

The proposed method is trained on ImageNet, CC3M and CC12M and compared against one supervised learning method and various SSL methodologies. It is shown that $\mathbb{X}$-CLR leads to consistent improvements for classification and background decomposition compared to the baselines, with minimal increase in computation.

**Strengths:**

This paper recognizes an important problem, which is that commonly used contrastive objectives assume binary similarity scores, which in many settings do not make much sense. The proposed method is an intuitive way of generalizing contrastive learning to non-binary similarity scores, effectively dealing with this problem. $\mathbb{X}$-CLR is tested through a comprehensive suite of experiments, showcasing its effectiveness and highlighting some of the important design choices that should be taken into account when using this objective.

The paper is well written with a clear explanation of the problem of existing contrastive learning methods, followed by a great presentation of the proposed method. Moreover, the experiment descriptions are both clear and concise, effectively highlighting the strengths and weaknesses of the method. Finally, the paper is well structured overall, making it enjoyable to read.

**Weaknesses:**

In my opinion, there are no significant weaknesses in the paper. The only thing I was wondering regarding the experiments is how fair the comparison between $\mathbb{X}$-CLR and the SSL methods in Table 1 is. It seems to me that $\mathbb{X}$-CLR is a supervised method in this setting. If the authors agree with me, then it might be nice to explicitly mention this in the text to avoid any possible confusion. That being said, I do think including the comparison is important to showcase the effect of proper similarity scoring, so this is a minor remark.

Aside from that, I have some small notes regarding the presentation, which I will list below:
  - It may be useful to shortly mention what $f_\theta$ and $K$ are after equation (1) in line 202 if space is not a concern. It is reasonably clear from the context, but being explicit rarely hurts.
  - In the data  scarcity setting (figure 3a), SupCon appears to be better than $\mathbb{X}$-CLR, while the text mentions that $\mathbb{X}$-CLR improves over SupCon.
  - The coloring of SupCon and SimCLR in figures 3a and 3b is flipped, which is a bit confusing at first glance.
  - The paragraph related to table 4 does not mention training on CC3M, which might be useful to mention for clarity.

**Questions:**

As mentioned before, I think the given experiments paint a clear picture of the method. So the questions listed here are mostly out of interest and not things that I necessarily think make sense to include in the paper. That being said, I was wondering a few things about the method:
  - What happens in imbalanced settings? I imagine that the similarities for an instance with rare caption/label/meta data would be much lower than for instances with more common caption/label/meta data. Do you think this would have a strong effect on the method? Have you tried experimenting with for example ImageNet-LT?
  - An annoying problem of contrastive learning is its sensitivity to the batch size ($B$), which stems from its effect on the ratio of positives to negatives across the training set ($B^{-1}$). I imagine that $\mathbb{X}$-CLR partially solves this problem, especially for larger batch sizes, given that the similarities $\mathbf{G}_{ij}^{(\text{soft})}$ approximately follow $\mathbb{P}\_X(\text{sim}(X\_i, X))$, where $\mathbb{P}\_X$ denotes the population distribution on inputs $X$. The only bias being the small effect of the true positive $\mathbf{G}\_{ii}^{(\text{soft})} = 1$. Is this something you looked into? If yes, how sensitive is $\mathbb{X}$-CLR to changes in the batch size? Furthermore, does the application of the softmax to obtain the distribution $s$ have any negative effects for larger batch sizes (since $s_i \rightarrow 0$ when $B \rightarrow \infty$)?
  - Do the similarities for ImageNet (shown in figure 5a) come from the way the text encoder input is constructed (so from the fixed template part)? If this is the case, do the similarities not give a false view of how strongly related images are? I expected there to be at least some very low similarities for each image, although I might be misunderstanding something here.

---

> ### Author Response · Authors · 2024-11-19
> **Response**
>
> Thank you for your feedback! We are glad you find this paper well written and enjoyable to read.
> Below, we address your questions and concerns. If you have any additional questions, we remain available to answer them.
>
> ### Notes regarding presentation.
>
> > **It may be useful to shortly mention what $f_\theta$ and $K$ are after equation (1) in line 202 if space is not a concern.**
>
> Thank you for pointing out this omission. We added this to the main text in the revision.
>
> > **In the data scarcity setting (figure 3a), SupCon appears to be better than X-CLR, while the text mentions that X-CLR improves over SupCon.**
>
> Thank you for pointing this out, we have corrected the wording in the main text in the revision. SupCon in this case indeed is better than X-CLR when trained with a subset of the data.
>
> > **The coloring of SupCon and SimCLR in figures 3a and 3b is flipped, which is a bit confusing at first glance. **
>
> Thank you for pointing this out. We have fixed the coloring to make it consistent across the two plots in the revision.
>
> > **The paragraph related to table 4 does not mention training on CC3M, which might be useful to mention for clarity.**
>
> Thank you for noticing this, we fixed this omission in the revision.
>
> ###  Questions
>
> > **What happens in imbalanced settings? Do you think this would have a strong effect on the method? Have you tried experimenting with for example ImageNet-LT?**
>
> We haven't experimented with explicitly imbalanced settings, but we can see our experiments with conceptual captions as imbalanced data. In that dataset, we see that the similarities are much more skewed towards 0, as shown in figure 5a. When looking at the caption data in CC3M, we see that for example the word 'actor' occurs in ~125K captions, while the word 'athlete' occurs only in 18K samples, meaning that actor images will get more signal on average. Therefore, the data is already quite imbalanced, with athlete images coming up a lot less than actor images. In table 6, we see that although our method outperforms CLIP when trained on CC12M, better balanced ImageNet labels yield better representations. This leads us to the hypothesis that imbalanced data could be an issue when applying our method, although we believe that this issue can be addressed with custom data sampling. For example, to have more learning signal, we can construct the batch in a way that there are always some samples that are similar between each other, addressing the imbalance. This would be akin to hard negative sampling in contrastive learning (https://arxiv.org/pdf/2010.04592). We haven't investigated this in detail, but this could be an interesting direction for future work.
>
> > **An annoying problem of contrastive learning is its sensitivity to the batch size (B)... I imagine that X-CLR partially solves this problem, especially for larger batch sizes... Is this something you looked into? If yes, how sensitive is X-CLR to changes in the batch size?**
>
> In my understanding, contrastive learning usually requires a large batch size to sample enough diverse negatives to train a good representations (e.g. Figure 9 in SimCLR paper) . We hypothesize that X-CLR can extract more signal per batch due to soft similarities, and therefore doesn't require batch sizes to be that large. On the other hand, when the batch size is large, as you correctly noted, the weights of individual samples similarities will go down. However, like with contrastive learning, it's possible that having a bigger batch will help due to having more samples' pairs in each batch. We do note that change in batch size may require adjusting the parameter $\tau_s$, as we show in Figure 5b.
>
> We didn't extensively experiment with various batch sizes, and used the fixed batch size dictated by the hardware capacity. It would be interesting to see how X-CLR performs with different batch sizes, but we leave this study to future work.
>
> > **Do the similarities for ImageNet (shown in figure 5a) come from the way the text encoder input is constructed (so from the fixed template part)? If this is the case, do the similarities not give a false view of how strongly related images are?**
>
> Yes, your understanding is right. The histograms show actual values of $s_{i,j}$, i.e. values after the softmax with temperature $\tau_s$. The non-zero similarities are due to the fact that the template is shared, and therefore each caption has a shared part "a photo of a". We agree that this is not ideal, but our results show that although the similarities for ImageNet are almost never zero, the signal from relative similarities still helps the model learn better representations.
> We tried randomizing the templates used to generate captions and calculating similarity between samples using the average over a set of templates, but didn't see any improvement over just using a fixed "a photo of a _ " template.

---

> > ### Comment · Reviewer_R31V · 2024-11-25
> >
> > I thank the authors for answering my questions and addressing my concern in the revised manuscript. I will maintain my rating.

---

> > > ### Author Response · Authors · 2024-11-25
> > >
> > > Thank you for taking the time to read the paper and for your questions and feedback!

---

### Official Review · Reviewer_QWUr · 2024-11-01

**Soundness:** 2
**Presentation:** 2
**Contribution:** 2
**Rating:** 5
**Confidence:** 5

**Summary:**

This paper proposes a new contrastive objective that uses a target similarity graph based on additional information or models. The modification is applied to contrastive losses in both multi-modal learning (image-text CLIP training) as well as self-supervised learning (images). The proposed modification is called “X-Sample Contrastive Loss” or X-CLR, where the model assumes access to an external source of metadata and possibly pretrained models that can define a graph of similarity between inputs. The goal is to replace the 0-1 hard similarity graph used in contrastive methods with soft similarity graphs based on some ground-truth information.

**Strengths:**

- Unifying SSL, multi-modal contrastive learning, and distillation methods in one formulation is helpful in defining generalized training methods.
- The ablation on the similarity sources in Table 2 is interesting as it shows strong text encoders are not helpful for X-CLR loss than more simple sentence transformer.

**Weaknesses:**

- The paper should clearly separate the SSL methods on images from image-text methods such as CLIP and clearly define X-CLR modification for both setups. There is a clear difference between contrastive losses for the two setups, one relies on image augmentations to define contrastive pairs while the other relies on paired image-text datasets. Figure 1.b presents a pseudocode for the X-CLR loss that is valid only when paired image-text data is available. In fact, the only reason this loss does not collapse to a trivial and all equal solution is that the `text_emb` is detached from the computation graph.
- The tables and comparisons should specify which models have access to additional data and/or additional pretrained models. For example, Table 1 compares X-CLR to SimCLR while X-CLR not only has access to the labels, but also access to pretrained text models. The comparison to methods that do not have the advantages of X-CLR is not fair. Specifically, when the method using the labels in ImageNet to define the loss, it can no longer be fairly compared with SSL methods that do not have access to labels. For example, the analysis in section 4.5 that compares the overhead of X-CLR to SimCLR needs to account for the training time of the text encoder as well.
- On line 354, the paper claims that “X-CLR is more general than model distillation methods as the similarities can come from any source” and cites the TinyCLIP [1]. Are the authors claiming X-CLR can be better than CLIP distillation methods? If so, the paper should compare to state-of-the-art CLIP distillation methods such TinyCLIP [1] and MobileCLIP [2]. That being said, I did not find a clear definition of X-CLR for image-text loss that allows for training both image/text towers even if it uses pretrained image and/or text pretrained models. Specifically, the loss should have image2text and text2image terms or show that it can learn a comparable representation without these terms.

1. Wu et al, Tinyclip: Clip distillation via affinity mimicking and weight inheritance, ICCV 2023.
2. Vasu et al., MobileCLIP: Fast Image-Text Models through Multi-Modal Reinforced Training, CVPR 2024.

**Questions:**

- Are both Tables 1,3,4 using pretrained frozen sentence transformers as the text encoder?

---

> ### Author Response · Authors · 2024-11-19
> **Response**
>
> Thank you for your feedback! Below, we address your questions and concerns. We remain available to answer any additional questions you may have. If we addressed your questions well, we kindly ask you to take that into account when reconsidering the evaluation of this paper.
>
> Responses:
>
> > **The paper should clearly separate the SSL methods on images from image-text methods such as CLIP and clearly define X-CLR modification for both setups.**
>
> We apologize for the confusion. We compare to other SSL methods on images because our method is based on SimCLR objective and introduces additional information to the loss. This makes SimCLR a natural comparison to analyze the effect of introducing extra similarities to the objective. In the section where we use Conceptual Captions data, we compare to CLIP because in our case we use text captions to build the similarity matrix. CLIP is then a natural comparison because it also uses both images and text during training.
>
> We believe the difference from SimCLR and SupCon is explained in Section 3 and appendix 9. We additionally add a section in the appendix to discuss the difference from CLIP, see appendix 10 in the revision. We hope this clarifies this issue enough.
>
> > **Figure 1.b presents a pseudocode for the X-CLR loss that is valid only when paired image-text data is available.**
>
> Thank you for pointing this out. We chose to show this pseudocode because that's the method we ended up using the most throughout the paper. That said, using a text encoder to calculate similarities is only one way to obtain the target similarities. To address your concern, we additionally introduce pseudocode for general similarities in the Appendix 11 in the revision. We also copy it here:
>
> ```
> def training_step(images, sims):
>     # encode images with the model we are training
>     img_emb = img_enc(images)  # N x D_img
>     # convert targets to distributions
>     tgt_sims = softmax(sims / tao_s, dim=1)
>     # calculate similarities
>     img_sims =
>         softmax(img_emb @ img_emb.T / tao, dim=1)
>     # the loss is the cross-entropy
>     return CE(tgt_sims, img_sims, dim=1).mean()
> ```
>
> `sims` here comes from the the target similarities graph. As you can see, neither the text encoder nor captions come up here at all.
>
> > **The tables and comparisons should specify which models have access to additional data and/or additional pretrained models.**
>
> Thank you for pointing this out. We apologize for the confusion. To clarify the distinction, we modified Table 1 in the revision to clearly show which methods use labels. Please let us know if this addresses your concerns.
>
> > **The analysis in section 4.5 that compares the overhead of X-CLR to SimCLR needs to account for the training time of the text encoder as well.**
>
> We agree that training the text encoder has a hidden computational cost. However, we choose not to include training time of the text encoder in this table because the similarities may be computed in a variety of ways depending on the data available, and using the text encoder is only one of them. We acknowledge that obtaining the similarities may be computationally expensive. However, in this table, we only aim to highlight that incorporating additional information into the loss function doesn't make it significantly slower.
>
> > **Are the authors claiming X-CLR can be better than CLIP distillation methods?.**
>
> We apologize for the confusion, we are not claiming that our method is better than CLIP distillation methods. Our message is that our method is not focusing on distillation, but on introducing similarities coming from an external source, whatever that source may be. We experimented with a variety of similarities sources, including WordNet hierarchy distance which doesn't involve running another model at all. Having those similarities come from an already trained model (distillation) is just one approach. In that sense, our approach encompasses distillation.
>
> > **I did not find a clear definition of X-CLR for image-text loss that allows for training both image/text towers even if it uses pretrained image and/or text pretrained models.**
>
> As you correctly pointed out, X-CLR doesn't introduce a cross-modality loss. In this work, we focus on training the image encoder by introducing cross-sample similarities. Using a text encoder is just one way of computing those similarities. As we show in Table 2 in the paper, similarities can come purely from the data, e.g. WordNet hierarchy distance.
> This is an interesting direction of research, but we leave the extension of X-CLR to training both image and text towers to future work.
>
> > **Are both Tables 1,3,4 using pretrained frozen sentence transformers as the text encoder?**
>
> Yes, that understanding is correct. Unless otherwise specified, the experiments throughout the paper use frozen sentence transformer.

---

> > ### Author Response · Authors · 2024-11-25
> >
> > Dear reviewer QWUr,
> >
> > We would like to remind you that the the end of the discussion period is in less than 2 days. We would appreciate it if you could go over our response when you get a chance. We believe we addressed your questions and concerns, but if you have further questions, we remain available to answer them. We kindly request that you take the response into account when reconsidering the score of the paper.

---

> > ### Comment · Reviewer_QWUr · 2024-11-25
> >
> > I thank authors for their responses and changes to the paper based on them. However, similar to Reviewer HZRT, I still have concerns about the contributions of this work. Particularly, the fact that the authors still include a claim about encompassing distillation is concerning. To me, the formulation is general but that's not a novelty. Rather, the novelty would come from interesting and novel choices for the source of metadata as well as exploring the effectiveness for training various models. Unfortunately, the paper is lacking in this aspect and the applied changes do not adequately address that. As such, I retain my rating.

---

> ### Author Response · Authors · 2024-11-25
>
> Thank you for taking the time to read our response!
>
> We have updated the manuscript and replaced the claim that "X-CLR is more general that distillation methods" with "X-CLR is not a model distillation method". We apologize for this overreaching claim, and hope that this addresses your concern. We do not claim to be better than model distillation, we only aim to emphasize that we use similarities that can come from any source, not necessarily another model.
>
> Thank you for the feedback regarding novelty and experiments. **We did investigate how X-CLR affects training on different datasets (Tables 1, 3, and 4), and how the choice of the metadata affects the performance (Table 2), and, according to reviewers HZRT and R31V, these experiments are comprehensive**. We will consider extending these experiments. In the meantime, please don't hesitate to share with us if you have any ideas regarding what experiments could strengthen this paper.

---

### Official Review · Reviewer_HZRT · 2024-11-02

**Soundness:** 2
**Presentation:** 2
**Contribution:** 3
**Rating:** 5
**Confidence:** 4

**Summary:**

This work presents X-CLR, a contrastive loss objective that replaces the usual binary positive/negative samples in contrastive learning with soft labels that can vary inside [0,1]. The resulting loss function requires pairwise similarity labels, which the authors suggest can come in the form of metadata or via text embeddings of captions generated from class labels. A comprehensive set of experiments on ImageNet and CC3M/CC12M show improvement in performance of X-CLR relative to existing contrastive learning baselines on classification tasks.

**Strengths:**

1) The proposed method and its advantages are intuitive and easy to understand, and there is a comparison with existing vision-based contrastive loss functions.

2) The suite of experimental results is comprehensive, and considers many useful ablations. I particularly appreciated the practical experiments about finetuning and refining existing vision backbones, and the explorations of high-quality vs noisy similarity data.

**Weaknesses:**

1) There is limited novelty to the proposed contrastive learning loss function, which is a relatively simple swapping of binary positive/negative sample labels in existing contrastive learning losses with a soft label.

2) Pursuant to the first point, there not much detailed analysis of how this method related to existing ones beyond a comparison of the binary vs. soft label graph.

3) The presentation of the paper is inconsistent - there are some very concise and informative figures, like Figure 1, but for example, there are many wording issues throughout, and there are other places where more detail is needed (this is particularly true for table captions).

4) Some weaknesses in experimental results - there is only one network architecture tested, though I accept the given justification to some extent.

**Questions:**

1) Some of the results you get for competing methods are far below what those methods report in their papers. Just as an example, in Table 1, ImageNet accuracy from your experiment with SimCLR is 63.4% vs. the 69.3% reported in their paper, and SupCon is 74.3% in your table vs. 78.7% reported in their paper. This is a very significant difference, since in the case of SupCon, their reported result outperforms X-CLR. Did you look into the large differences here? Can you explain them?

2) Please add some more detail to the table captions. The captions right now generally tell me what the takeaway should be, but they don't tell me the specifics of what I'm looking at: things like what the numbers actually represent (accuracy), what the columns mean (i.e., for Table 1, "Accuracy of networks pretrained on ImageNet with various contrastive learning algorithms, and evaluated on a set of..."), etc. Also, I think it is a stretch to call Theorem 1 a theorem - this is an re-expression of these losses in matrix form, and the cited paper doesn't even offer a proof. I think these can simply be stated.

3) I found the CLIP experiments section confusing. In your CLIP baseline, are you actually training a vision encoder and language encoder jointly with a CLIP loss, or are you taking a pretrained CLIP model and performing training with X-CLR loss? Also, in general I thought a deeper analysis with CLIP was missing from Section 3, particularly in the case of generating similarities with captions + a CLIP text encoder. Is X-CLR equivalent to CLIP training with a frozen / pre-trained text encoder in this case? If not, how is it different? I think this comparison is important to contextualize your method.

4) In the case of ImageNet, you use class labels to generate text captions, and then embed these to get text embeddings, which ultimately determine the connection strengths in the graph. This creates a similarity matrix that has some particular properties, i.e., images within the same class have a similarity of 1, as in the Supervised graph in Figure 1, along with other nonzero entries based on the similarities of two different images' classes. Does this end up being the same as SupCon? There are some similarities with supervised learning here, but there is no analysis of this similarity.

---

> ### Author Response · Authors · 2024-11-19
> **Response**
>
> Thank you for your feedback! Below, we address your questions and concerns. We remain available to answer any additional questions you may have. If we addressed your questions well, we kindly ask you to take that into account when reconsidering the evaluation of this paper.
>
> Responses:
>
> > **There is limited novelty to the proposed contrastive learning loss function...**
>
> In this work, we focus on introducing similarities into self-supervised learning. We show that introducing similarities during pre-training improves representation quality. We see the loss function itself as a technical detail. While we agree that the loss-function by itself has limited novelty, the introduction of external similarities to the objective is, to the best of our knowledge, novel.
>
> > **There not much detailed analysis of how this method related to existing ones beyond a comparison of the binary vs. soft label graph.**
>
> Thank you for pointing this out. We present an analysis comparing it to SupCon in Appendix A9.
> Additionally, we add a section to compare to CLIP in Appendix A10. Let us know if you have any other questions or concerns, or have concerns about comparisons to any other methods.
>
> > **There is only one network architecture tested**
>
> We agree this is a limitation, and it would be valuable to have results for e.g. transformer based encoders. We chose to only use ResNet-50 as it's the most studied architecture in SSL community, and allows for faster iteration due to its size. We unfortunately do not currently have the computational resources to conduct experiments with bigger models, and leave this extension to future work.
>
> **Concerns regarding table captions and writing**
>
> Thank you for pointing this out. We have revised the table captions to include information about the conducted experiments. Please let us know if this addresses your concerns, or if you have other feedback on writing.
>
> > **Some of the results you get for competing methods are far below what those methods report in their papers.**
>
> We apologize for this confusion. This difference you noted is due to the fact that we train all methods for 100 epochs. We acknowledge that SupCon benefits from significantly longer training (500 epochs), but train all methods for 100 epochs to due to computational constraints. We also mention this in the paper in appendix A7. We agree that this is a major detail, so it should be in the main text. To make this more clear, we added this to the main text in the revision.
>
> > **In your CLIP baseline, are you actually training a vision encoder and language encoder jointly with a CLIP loss, or are you taking a pretrained CLIP model and performing training with X-CLR loss?**
>
> CLIP baseline is just a CLIP model trained with the CLIP loss throughout, we do not fine-tune it in any way. To compare to our method, we only take the clip encoder, and perform linear probing on top of the frozen encoder.
>
> The only place where we used a pre-trained CLIP model is in our experiment with different similarity sources in table 2. There, we use a pre-trained CLIP text encoder to calculate target similarities for X-CLR.
>
> > **Is X-CLR equivalent to CLIP training with a frozen / pre-trained text encoder in this case?**
>
> X-CLR with CLIP encoder similarities is still not equivalent to CLIP training with a frozen encoder in this case. CLIP loss uses contrastive learning, and pushes together positive pairs of text and image representations, while pushing the rest apart. X-CLR loss differs in two key aspects:
> - During training, unlike CLIP, X-CLR does not push image representations to be similar to text representations. In our method, we do not assume we have access to text representations, we only assume access to sample similarity values. We work with similarities between image representations, while clip works with image-text similarities.
> - CLIP loss is not 'soft', i.e. the similarities in the contrastive loss are 0 or 1.
>
> We include a more detailed analysis in the appendix in the revision, see Appendix 10.
>
> > **Does the similarity matrix end up being the same as SupCon?**
>
> The similarity matrices of SupCon and X-CLR are indeed similar, but are not identical. We show a more detailed analysis of this in appendix A.9. For ImageNet, as you have correctly pointed out, similarities between the samples of the same class are going to be 1 like in SupCon, but for samples of other classes, the similarities can be between 0 and 1. The exact amount of weight put on the positives coming from the same class is controlled by the softmax temperature parameter $\tau_s$, with the X-CLR loss becoming SupCon as $\tau_s \rightarrow \infty$. With lower temperature values, more weight will be put on the samples other than the true positive, see Figure 5b in the paper.

---

> > ### Author Response · Authors · 2024-11-25
> >
> > Dear reviewer HZRT,
> >
> > We would like to remind you that the the end of the discussion period is in less than 2 days. We would really appreciate it if you could go over our response. We believe we addressed your questions and concerns, but if you have further questions, we remain available to answer them. We kindly request that you take the response into the account when reconsidering the score of the paper.

---

> > > ### Comment · Reviewer_HZRT · 2024-11-27
> > >
> > > I appreciate the authors' response to my review. To respond specifically to a couple main points:
> > >
> > > - Novelty: I still find that the novelty of the method is limited. I sincerely don't mean to be dismissive when I say this, but I mean this both in the sense of the loss function, but also, in a broader sense I think the addition of the similarity matrix is a relatively straightforward idea, and I didn't find anything in the execution of the idea that I wouldn't expect.
> > > - Experimental discrepancies & other architectures: I understand that this 100 epochs vs 500 epochs distinction is the cause for different results. But I think training for 100 epochs on these tasks is likely significantly underfitting (as indicated by your results on SupCon & SimCLR vs their papers'), and it's simply not enough to get a real understanding of how the method performs in a practical setting. When it comes down to it I'm not confident that, e.g., your method would actually outperform SupCon in a longer experiment. I am very sympathetic to compute constraints, but in this case, I think given your available compute, you should have prioritized a few (for example) 300 epoch experimental runs, even if that meant not carrying out some of your other supporting experiments.
> > > - CLIP loss similarity: I understand that X-CLR works with image-image similarity rather than the image-text similarity of a CLIP loss. You have pointed out that text representations are not required for your method, but I was referring to the situation in which you generate the image-image similarity matrix *using* text representations, as is the norm in most of the paper. If you generate the similarity matrix in this way, then you are implicitly pushing image representations together or apart based on text representations; this seems like has some similarity to CLIP training with a frozen text encoder, since in that case you are just, e.g., pushing images together by pushing them toward the same fixed text vectors in latent space. I agree that this is probably not mathematically identical, but I think there is more of a quantifiable relationship between these two than what you've responded with or put in Appendix A.10.
> > >
> > > Due to the above points, I will retain my rating in my review.

---

> ### Author Response · Authors · 2024-11-29
>
> Thank you for your response! We respond to the points in your response below:
>
> ## Novelty
> We agree that the idea of the loss function is intuitive and straightforward, but we consider this a strength rather than a weakness. Although a similar objective has been used for distillation, using it to inject additional similarity information to the learning objective is, to the best of our knowledge, novel, and **we don't think the simplicity of the idea detracts from the novelty.**
>
> ## Experiments
>  We agree that running ImageNet training for more epochs would be a great addition to the paper, we will do our best to add these results, but we don't have time to do this before the end of the discussion period unfortunately. That said, we disagree that running on ImageNet for more epochs is more important than other experiments we ran. In this work, we focused on extending the method to bigger datasets as opposed to focusing solely on ImageNet as was done in SupCon and SimCLR papers. **We believe that experiments scaling our method to Conceptual Captions dataset with 12M samples in table 3 highlight scalability of our method to a more practical setting with longer training and a bigger dataset better than training for more epochs on ImageNet would.** Running experiments on the full Conceptual Captions dataset with 12 million samples (CC12M) took 9 days for a single run, and took significant computational resources. We kept the number of epochs fixed, and compared to the ImageNet dataset with 1M samples, our experiments on CC12M amount to 12 times more gradient updates than experiments on ImageNet with 100 epochs.
>
> ##  CLIP loss similarity
> This is a good point, we thought about this connection when building this method. If we use text similarity as is done in most of the experiments, we do not push image representations to be similar to text representations as is done in CLIP.  As a result, **the only thing carried from the text representations to image representations when training with X-CLR is the relationship between samples. The representations themselves can be different in terms of at least rotation and reflection.** For example, let's define representations learned by X-CLR as $X \in \mathbb{R}^{N \times D}$, where N is the number of data samples, and D is the representation dimension. Then the pairwise similarities are:
> $$S = XX^\top$$
> If we perfectly optimize the X-CLR objective we learn representations with similarities that are the same as those of a frozen text encoder. For the frozen text encoder representations $Y \in \mathbb{R}^{N \times D}$, we'd get $S_\mathrm{text} = YY^\top$. So, for the perfectly optimized objective, we'd get $S = S_\mathrm{text}$. But that doesn't mean that $X=Y$ as would be the case for using the CLIP objective. To show this, let us introduce an orthonormal rotation/reflection matrix $T \in \mathbb{R}^{D \times D}$, such that  $T^{-1}=T^\top$. Then, if we rotate representations $X'=XT$, we get the similarity:
> $$S' = X'X'^\top=XT(XT)^\top=XTT^\top X^\top = XX^\top = S$$
>
> This means that rotating and/or reflecting the representations doesn't change the similarities.
> Therefore minimizing X-CLR objective learns image representations that could be a rotation/reflection of the text representations.
>
> **We will add this explanation to appendix A.10**. Unfortunately, we are no longer able to update the revision.
>
>
> Thank you again for the insightful feedback! We remain available to answer any other questions you may have.

---

> > ### Author Response · Authors · 2024-12-02
> >
> > Dear reviewer,
> >
> > We would like to remind you that the the end of the discussion period is in less than 24 hours. We would really appreciate it if you could go over our response and let us know if it addresses your concerns.

---

### Author Response · Authors · 2024-11-19
**General response**

We thank all the reviewers for taking the time to read our submission and providing the feedback!

We are glad that the reviewers found:
- that the problem we investigate is **very important in contrastive learning literature** (reviewer R31V)
- our method **intuitive and easy to understand** (reviewers HZRT, R31V)
- the **experiments comprehensive** and interesting (reviewers HZRT, QWUr, R31V)
- the overall paper **well-written** (reviewers R31V, MXJi).

# Revision updates
We have and updated the paper and uploaded the revision with the following changes based on the provided feedback:
- Revised table captions to make results more clear (reviewer HZRT)
- Added columns to distinguish methods that use labels to table 1 to clearly show what data the learning methods have access to (reviewers QWUr and R31V).
- Fixed figure colors in figure 3a and 3b (reviewer R31V).
- Added a section in the appendix comparing our method to CLIP to address any confusion regarding the relationship between X-CLR and CLIP objectives (Appendix 10) (reviewers HZRT and QWUr)
- Added pseudocode to show X-CLR loss in the general case, when the similarities don't come form the text encoder (Appendix 11) (reviewer QWUr)
- Mentioned that we train all methods for 100 epochs and do not explore training longer as was done in SimCLR and SupCon papers in the main text (reviewer HZRT). As mentioned in the response, we believe CC12M experiments are more valuable than training for more epochs on ImageNet as they test X-CLR in a more real-world setting with noisy labels.

# Main concerns
- Lack of novelty in the objective (reviewers HZRT and QWUr) - While we agree that the extension of this loss function to soft similarities is simple and is akin to the distillation objectives, to the best of our knowledge, **using such a loss function to inject external similarity information into the objective is novel. Moreover, we believe simplicity does not detract from the novelty.** We also believe this is an important contribution because, as reviewer R31V pointed out, **we consider an important problem in contrastive learning objectives.**
- Experiments not supporting the method enough (reviewers HZRT and QWUr) - Our experiments explore different choices of the similarity graph (table 2), comparison to a wide range of baslines on ImageNet (table 1), fine-tuning performance (table 4), extension to the large conceptual captions dataset with 12M samples (table 3), KNN evaluations, temperature ablations and low-data regime performance (Figure 3). **We believe our experiments thoroughly test this method in a variety of settings, with different datasets, benchmarks, and probing methods. Reviewers HZRT QWUr and R31V agree that the experiments are comprehensive.**

We hope the improved revision addresses the concerns you have. We remain available to answer any additional questions you may have. Thank you for helping us make this paper better!

---

### Meta-Review · Area_Chair_VfbZ · 2024-12-20

**Metareview:**

This submission proposes X-CLR, a contrastive learning objective that incorporates soft similarity labels derived from external metadata or pretrained models, as opposed to traditional binary similarity scores. While the method shows some performance improvements over baseline methods in classification tasks, there are critical concerns raised by reviewers regarding the novelty, fairness of comparisons, and the robustness of the experiments.

The proposed method offers a practical solution to a well-recognized problem in contrastive learning, with demonstrated performance improvements. While some reviewers remained skeptical of novelty, they acknowledged the potential value of the approach for future research.

**Additional Comments On Reviewer Discussion:**

The authors provided detailed responses, addressing concerns about comparisons, novelty, and reliance on metadata. They revised the manuscript to clarify claims, improve presentation, and add pseudocode for general applicability. While their rebuttal addressed some technical concerns, it did not fully alleviate novelty-related reservations for some reviewers. The authors are strongly encouraged to very carefully clarify the concerns in the final paper.

---

### Decision · Program_Chairs · 2025-01-22

Accept (Poster)